# Doubly Optimal Policy Evaluation for Reinforcement Learning

**Shuze Daniel Liu**
Department of Computer Science
University of Virginia
`shuzeliu@virginia.edu`

**Claire Chen**
School of Arts and Science
University of Virginia
`clairechen@email.virginia.edu`

**Shangtong Zhang**
Department of Computer Science
University of Virginia
`shangtong@virginia.edu`

## Abstract

Policy evaluation estimates the performance of a policy by (1) collecting data from the environment and (2) processing raw data into a meaningful estimate. Due to the sequential nature of reinforcement learning, any improper data-collecting policy or data-processing method substantially deteriorates the variance of evaluation results over long time steps. Thus, policy evaluation often suffers from large variance and requires massive data to achieve the desired accuracy. In this work, we design an optimal combination of data-collecting policy and data-processing baseline. Theoretically, we prove our doubly optimal policy evaluation method is unbiased and guaranteed to have lower variance than previously best-performing methods. Empirically, compared with previous works, we show our method reduces variance substantially and achieves superior empirical performance.

## 1 Introduction

Reinforcement learning (RL, Sutton and Barto (2018)) has achieved remarkable success in various sequential decision-making problems. For example, RL algorithms have reduced energy consumption for Google data centers' cooling by $40\%$ (Chervonyi et al., 2022), predicted protein structures with competitive accuracy (Jumper et al., 2021), and discovered faster matrix multiplication algorithms (Fawzi et al., 2022). When applying RL algorithms, *policy evaluation* plays a critical role to allow practitioners to estimate the performance of a policy before committing to its full deployment and test different algorithmic choices. A commonly used approach among RL practitioners for policy evaluation is the on-policy Monte Carlo method, where a policy (i.e., the target policy) is evaluated by directly executing itself. However, using the target policy itself as the behavior policy is not optimal (Hanna et al., 2017; Liu and Zhang, 2024; Liu et al., 2024b; Chen et al., 2024), leading to evaluation with high variance. This suboptimality of on-policy evaluation results in extensive needs for collecting online samples to achieve a desired level of accuracy.

In many scenarios, heavily relying on online data is not preferable, since collecting massive online data through real-world interaction is both expensive and slow (Li, 2019; Zhang, 2023). Even with a well-developed simulator, complex tasks like data center cooling take 10 seconds per step (Chervonyi et al., 2022), making the evaluation of a policy requiring millions of steps prohibitively expensive. To address the expensive nature of online data, offline RL is proposed to mitigate the dependency on online data. However, there are often mismatches between the offline data distribution and the data distribution induced by the target policy, leading to uncontrolled and ineliminable bias (Jiang and Li, 2016; Farahmand and Szepesvári, 2011; Marivate, 2015). As a result, a policy with high performance on offline data may actually perform very poorly in real deployment (Levine, 2018). Consequently, both online and offline RL practitioners still heavily rely on online policy evaluation methods(Kalashnikov et al., 2018; Vinyals et al., 2019).

Improving the online sample efficiency for policy evaluation by reducing the variance of estimators is thus a critical challenge in the RL community. In this paper, we tackle this challenge by decomposing policy evaluation into two phases: data collecting and data processing. Our contributions are summarized as follows:

1. We design a doubly optimal estimator by proposing an **optimal data-collecting policy** and an **optimal data-processing baseline**. They are carefully tailored to each other to guarantee both unbiasedness and substantial variance reduction.

2. Theoretically, we prove our method has guaranteed lower variance than the on-policy Monte Carlo estimator, and is superior to previously best methods (Jiang and Li, 2016; Liu and Zhang, 2024). Moreover, such superiority grows over the time horizon as ensured by rigorous mathematical analysis.

3. Empirically, we show that our method reduces variance substantially compared with previous works and achieves state-of-the-art performance across a broad set of environments.

## 2    RELATED WORK

Reducing the variance for policy evaluation in reinforcement learning (RL) has been widely studied. One rising approach is variance reduction by designing a proper data-collecting policy, also known as the behavior policy. Noticing that the target policy itself is not the best behavior policy, Hanna et al. (2017) formulate the task of finding a variance-reduction behavior policy as an optimization problem. They use stochastic gradient descent to update a parameterized behavior policy. However, the stochastic method has been known to easily get stuck in highly suboptimal points in just moderately complex environments, where various local optimal points exist (Williams, 1992). *By contrast, our method directly learns the globally optimal behavior policy without doing a policy search.* Moreover, their method requires highly sensitive hyperparameter tuning to learn the behavior policy effectively. Specifically, the learning rate can vary by up to $10^5$ times across different environments, as reported in the experiments of Hanna et al. (2017). This extreme sensitivity requires online tuning, consuming massive online data. *By contrast, we propose an efficient algorithm to learn our optimal behavior policy with purely offline data.* Furthermore, Hanna et al. (2017) constrain the online data to be complete trajectories. *By contrast, our method copes well with incomplete offline data tuples, which is widely applicable.*

Zhong et al. (2022) also aim to reduce the variance of policy evaluation through designing a proper behavior policy. They propose adjusting the behavior policy to focus on under-sampled data segments. Nevertheless, their method necessitates complete offline trajectories generated by known policies and assumes a strong similarity between the behavior and target policies, limiting the generalizability. *By contrast, our method effectively uses incomplete offline segments from unknown and diverse behavior policies.* Moreover, the estimates made by Zhong et al. (2022) lack theoretical guarantees of unbiasedness nor consistency. *By contrast, we theoretically prove that our estimate is inherently unbiased.* Another approach by Mukherjee et al. (2022) investigates behavior policies aimed at reducing variance in per-decision importance sampling estimators. However, their results are limited to tree-structured MDPs, a significant limitation since most problems do not adhere to tree structure. *By contrast, our method works on general MDPs without any restriction on their inherent structures.* Moreover, Mukherjee et al. (2022) explicitly require the knowledge of transition probability and, therefore, suffer from all canonical challenges in model learning (Sutton, 1990; Sutton et al., 2008; Deisenroth and Rasmussen, 2011; Chua et al., 2018). *By contrast, our approach is model-free and can use off-the-shelf offline policy evaluation methods (e.g. Fitted Q-Evaluation, Le et al. (2019)).* The current state-of-the-art method in behavior policy design is proposed by Liu and Zhang (2024), where they find an optimal and offline-learnable behavior policy with the per-decision importance sampling estimator. However, all these approaches (Hanna et al., 2017; Mukherjee et al., 2022; Zhong et al., 2022; Liu and Zhang, 2024) only consider optimality in the data-collecting process, while ignoring the potential improvement in data-processing steps. *By contrast, we model the variance reduction as a bi-level optimization problem, where the behavior policy is optimized with a baseline function. This fundamental difference makes our method superior in a broader context, as theoretically and empirically demonstrated in Section 5 and Section 7.*

Besides behavior policy design, another popular approach for reducing the variance in policy evaluation is using the baseline functions. Jiang and Li (2016) propose a doubly robust estimator by

incorporating a baseline function into the plain per-decision importance sampling estimator. However, their method assumes that the behavior policy is fixed and given, but does not discuss how to choose a proper behavior policy. Ignoring the choice of behavior policy loses the opportunity to save online samples manyfold. *By contrast, our method achieves optimality in both the design of behavior policy and the choice of baseline, thus outperforming the estimator of Jiang and Li (2016) both theoretically and empirically.* Thomas and Brunskill (2016) extend the method of Jiang and Li (2016) into the infinite horizon setting, proposing a weighted doubly robust estimator. However, their method introduces bias into the estimator, potentially leading the estimation to systematically deviate from the true return of the target policy.

## 3 BACKGROUND

In this paper, we study a finite horizon Markov Decision Process (MDP, Puterman (2014)). In this MDP, there is a finite action space $\mathcal{A}$, a finite action space $\mathcal{A}$, a transition probability function $p : \mathcal{S} \times \mathcal{S} \times \mathcal{A} \to [0, 1]$, a reward function $r : \mathcal{S} \times \mathcal{A} \to \mathbb{R}$, an initial state distribution $p_0 : \mathcal{S} \to [0, 1]$, and a constant horizon length $T$. For simplifying notations, we consider the undiscounted setting without loss of generality. Our method naturally applies to the discounted setting as long as the horizon is fixed and finite (Puterman, 2014). We define a shorthand $[n] \doteq \{0, 1, \ldots, n\}$ for any integer $n$.

The MDP process begins at time step $0$, where an initial state $S_0$ is sampled from $p_0$. At each time step $t \in [T-1]$, an action $A_t$ is sampled based on $\pi_t(\cdot \mid S_t)$. Here, $\pi_t : \mathcal{A} \times \mathcal{S} \to [0, 1]$ denotes the policy at time step $t$. Then, a finite reward $R_{t+1} \doteq r(S_t, A_t)$ is given by the environment and a successor state $S_{t+1}$ is obtained based on $p(\cdot \mid S_t, A_t)$. We use abbreviations $\pi_{i:j} \doteq \{\pi_i, \pi_{i+1}, \ldots, \pi_j\}$ and $\pi \doteq \pi_{0:T-1}$. At each time step $t$, the return is defined as $G_t \doteq \sum_{i=t+1}^{T} R_i$. Then, we define the state-value and action-value functions as $v_{\pi,t}(s) \doteq \mathbb{E}_\pi [G_t \mid S_t = s]$ and $q_{\pi,t}(s, a) \doteq \mathbb{E}_\pi [G_t \mid S_t = s, A_t = a]$. We adopt the total rewards performance metric (Puterman, 2014) for the measurement of the performance of policy $\pi$, which is defined as $J(\pi) \doteq \sum_s p_0(s) v_{\pi,0}(s)$. In this work, we use Monte Carlo methods, as introduced by Kakutani (1945), for estimating the total rewards $J(\pi)$. The most straightforward Monte Carlo method is to draw samples of $J(\pi)$ through the online execution of the policy $\pi$. The empirical average of the sampled returns converges to $J(\pi)$ as the number of samples increases. Since this method estimates a policy by executing itself, it is called on-policy learning (Sutton 1988).

Moving forward, we focus on off-policy evaluation, where the goal is to estimate the total rewards $J(\pi)$ of an interested policy $\pi$, which is called the *target policy*. Data for off-policy evaluation are collected by executing a different policy $\mu$, called the *behavior policy*. In off-policy evaluation, we generate each trajectory $\{S_0, A_0, R_1, S_1, A_1, R_2, \ldots, S_{T-1}, A_{T-1}, R_T\}$ by a behavior policy $\mu$ with $A_t \sim \mu_t(\cdot|S_t)$. We use a shorthand $\tau_{t:T-1}^{\mu_{t:T-1}} \doteq \{S_t, A_t, R_{t+1}, \ldots, S_{T-1}, A_{T-1}, R_T\}$ for a trajectory generated by the behavior policy $\mu$ from time step $t$ to $T-1$ inclusively. We use the importance sampling ratio to reweight the rewards obtained by the behavior policy $\mu$, in order to give an estimate of $J(\pi)$. We define the importance sampling ratio at time step $t$ as $\rho_t \doteq \frac{\pi_t(A_t|S_t)}{\mu_t(A_t|S_t)}$. Then, the product of importance sampling ratios from time $t$ to $t' \geq t$ is defined as $\rho_{t:t'} \doteq \prod_{k=t}^{t'} \frac{\pi_k(A_k|S_k)}{\mu_k(A_k|S_k)}$. In off-policy learning, there are several ways to use the importance sampling ratios (Geweke, 1988; Hesterberg, 1995; Koller and Friedman, 2009; Thomas, 2015). In this paper, we investigate the per-decision importance sampling estimator (PDIS, Precup et al. (2000)) and leave the investigation of others for future work. We define the PDIS Monte Carlo estimator as $G^{\text{PDIS}}(\tau_{t:T-1}^{\mu_{t:T-1}}) \doteq \sum_{k=t}^{T-1} \rho_{t:k} R_{k+1}$, which can also be expressed recursively as

$$G^{\text{PDIS}}(\tau_{t:T-1}^{\mu_{t:T-1}}) = \begin{cases} \rho_t \left( R_{t+1} + G^{\text{PDIS}}(\tau_{t+1:T-1}^{\mu_{t+1:T-1}}) \right) & t \in [T-2], \\ \rho_t R_{t+1} & t = T-1. \end{cases}$$

Under the classic policy coverage assumption (Precup et al., 2000; Maei, 2011; Sutton et al., 2016; Zhang, 2022; Liu et al., 2024a) $\forall t, s, a, \mu_t(a|s) = 0 \implies \pi_t(a|s) = 0$, this off-policy estimator $G^{\text{PDIS}}(\tau_{0:T-1}^{\mu_{0:T-1}})$ provides an *unbiased* estimation for $J(\pi)$, i.e., $\mathbb{E}\left[G^{\text{PDIS}}(\tau_{0:T-1}^{\mu_{0:T-1}})\right] = J(\pi)$.

In off-policy evaluation, a notorious curse is that the importance sampling ratios can be extremely large, resulting in infinite variance (Sutton and Barto, 2018). Even with the PDIS method, this fundamental issue still remains if the behavior policy significantly differs from the target policy,

particularly when the behavior policy assigns very low probabilities to actions favored by the target policy. Moreover, such degeneration of important sampling ratios typically grows with the dimensions of state and action spaces as well as the time horizon (Levine et al., 2020). One way to control for the violation in important sampling ratios is to subtract a baseline from samples (Williams, 1992; Greensmith et al., 2004; Jiang and Li, 2016; Thomas and Brunskill, 2017). Using $b$ to denote an arbitrary baseline function, the PDIS estimator with baseline is defined as

$$G^b(\tau_{t:T-1}^{\mu_{t:T-1}}) = \begin{cases} \rho_t \left(R_{t+1} + G^b(\tau_{t+1:T-1}^{\mu_{t+1:T-1}}) - b_t(S_t, A_t)\right) + \bar{b}_t(S_t) & t \in [T-2], \\ \rho_t(R_{t+1} - b_t(S_t, A_t)) + \bar{b}_t(S_t) & t = T-1, \end{cases} \quad (1)$$

where

$$\bar{b}_t(S_t) \doteq \mathbb{E}_{A_t \sim \pi}[b_t(S_t, A_t)]. \quad (2)$$

The variance of (1) highly depends on the importance sampling ratio $\rho_t = \frac{\pi_t(A_t|S_t)}{\mu_t(A_t|S_t)}$ and the choice of baseline function $b$.

## 4 VARIANCE REDUCTION IN REINFORCEMENT LEARNING

We seek to reduce the variance $\mathbb{V}(G^b(\tau_{0:T-1}^{\mu_{0:T-1}}))$ by designing an optimal behavior policy and an optimal baseline function at the same time. We solve the bi-level optimization problem

$$\min_b \min_\mu \quad \mathbb{V}(G^b(\tau_{0:T-1}^{\mu_{0:T-1}})) \quad (3)$$

$$\text{s.t.} \quad \mathbb{E}\left[G^b(\tau_{0:T-1}^{\mu_{0:T-1}})\right] = J(\pi),$$

where the optimal behavior policy $\mu^*$ and the optimal baseline function $b^*$ are carefully tailored to each other to guarantee both unbiasedness and substantial variance reduction.

Our paper proceeds as follows. In Section 4, we solve this bi-level optimization problem in closed-form. In Section 5, we mathematically quantify the superiority in variance reduction of our designed optimal behavior policy and baseline function, in comparison with cutting-edge methods (Jiang and Li, 2016; Liu and Zhang, 2024). In Section 7, we empirically show that such doubly optimal design reduces the variance substantially compared with the on-policy Monte Carlo estimator and previously best methods (Jiang and Li, 2016; Liu and Zhang, 2024) in a broad set of environments.

To ensure that the off-policy estimator $G^b(\tau_{0:T-1}^{\mu_{0:T-1}})$ is unbiased, the classic reinforcement learning wisdom (Precup et al., 2000; Maei, 2011; Sutton et al., 2016; Zhang, 2022) requires that the behavior policy $\mu$ covers the target policy $\pi$. This means that they constraint $\mu$ to be in a set

$$\Lambda^- \doteq \{\mu \mid \forall t, s, a, \pi_t(a|s) \neq 0 \implies \mu_t(a|s) \neq 0\}$$
$$= \{\mu \mid \forall t, s, a, \mu_t(a|s) = 0 \implies \pi_t(a|s) = 0\},$$

which contains all policies that satisfy the policy coverage constraint in off-policy learning (Sutton and Barto 2018). By specifying the policy coverage constraint, the optimization problem (3) is reformulated as

$$\min_b \min_\mu \quad \mathbb{V}(G^b(\tau_{0:T-1}^{\mu_{0:T-1}})) \quad (4)$$

$$\text{s.t.} \quad \mu \in \Lambda^-.$$

In this paper, compared with the classic reinforcement learning literature, we enlarge the search space of $\mu$ from this set $\Lambda^-$ to a set $\Lambda$. To achieve a superior and reliable optimization solution, we require $\Lambda$ to have two properties.

1. (Broadness) $\Lambda$ must be broad enough such that it includes all policies satisfying the classic policy coverage constraint (Precup et al., 2000; Sutton and Barto, 2018). Formally,

$$\Lambda^- \subseteq \Lambda. \quad (5)$$

2. (Unbiasedness) Every behavior policy in $\Lambda$ must be well-behaved such that the data collected by it can be used by the off-policy estimator to achieve unbiased estimation for all state $s$ and time step $t$. Formally, $\forall \mu \in \Lambda$,

$$\forall t, \forall s, \quad \mathbb{E}\left[G^b(\tau_{t:T-1}^{\mu_{t:T-1}}) \mid S_t = s\right] = v_{\pi,t}(s). \quad (6)$$

The space $\Lambda$ that satisfies those two properties will be defined shortly. We now reformulate our bi-level optimization problem as

$$\min_b \min_\mu \quad \mathbb{V}(G^b(\tau_{0:T-1}^{\mu_{0:T-1}})) \tag{7}$$
$$\text{s.t.} \quad \mu \in \Lambda.$$

Compared with the classic approach (4), our bi-level optimization problem (7) searches for $\mu$ in a broader space $\Lambda$ such that $\Lambda^- \subseteq \Lambda$. Thus, the optimal solution of our optimization problem must be *superior* to the optimal solution of the optimization problem with the classic policy coverage constraint. To solve our bi-level optimization problem (7), we first give a closed-form solution of the inner optimization problem

$$\min_\mu \quad \mathbb{V}(G^b(\tau_{0:T-1}^{\mu_{0:T-1}})) \tag{8}$$
$$\text{s.t.} \quad \mu \in \Lambda$$

for any baseline function $b$. Notably, this baseline function $b$ does not need to be any kind of oracle. We design the optimal solution of (8) for this baseline function $b$ without requiring any property on $b$. Now, we decompose the variance of our off-policy estimator $G^b(\tau_{0:T-1}^{\mu_{0:T-1}})$. By the law of total variance, $\forall b, \forall \mu \in \Lambda$,

$$\mathbb{V}\left(G^b(\tau_{0:T-1}^{\mu_{0:T-1}})\right)$$
$$=\mathbb{E}_{S_0}\left[\mathbb{V}\left(G^b(\tau_{0:T-1}^{\mu_{0:T-1}}) \mid S_0\right)\right] + \mathbb{V}_{S_0}\left(\mathbb{E}\left[G^b(\tau_{0:T-1}^{\mu_{0:T-1}}) \mid S_0\right]\right)$$
$$=\mathbb{E}_{S_0}\left[\mathbb{V}\left(G^b(\tau_{0:T-1}^{\mu_{0:T-1}}) \mid S_0\right)\right] + \mathbb{V}_{S_0}\left(v_{\pi,0}(S_0)\right). \tag{by (6)} \tag{9}$$

The second term in (9) is a constant given a target policy $\pi$ and is unrelated to the choice of $\mu$. In the first term, the expectation is taken over $S_0$ that is determined by the initial probability distribution $p_0$. Consequently, given any baseline function $b$, to solve the problem (8), it is sufficient to solve

$$\min_\mu \quad \mathbb{V}(G^b(\tau_{t:T-1}^{\mu_{t:T-1}}) \mid S_t = s) \tag{10}$$
$$\text{s.t.} \quad \mu \in \Lambda$$

for all $s$ and $t$. If we can find one optimal behavior policy $\mu^*$ that simultaneously solves the optimization problem (10) on all states and time steps, $\mu^*$ is also the optimal solution for the optimization problem (8). Denote the variance of the state value function for the next state given the current state-action pair as $\nu_{\pi,t}(s,a)$. Recall the notation $[T-2]$ is a shorthand for the set $\{0, 1, \ldots, T-2\}$. We have $\nu_{\pi,t}(s,a) \doteq 0$ for $t = T-1$, and $\forall t \in [T-2]$,

$$\nu_{\pi,t}(s,a) \doteq \mathbb{V}_{S_{t+1}}\left(v_{\pi,t+1}(S_{t+1}) \mid S_t = s, A_t = a\right). \tag{11}$$

Given any baseline function $b$, we construct a behavior policy $\mu^*$ as

$$\mu_t^*(a|s) \propto \pi_t(a|s)\sqrt{u_{\pi,t}(s,a)} \tag{12}$$

where $u_{\pi,t}(s,a) \doteq [q_{\pi,t}(s,a) - b_t(s,a)]^2$ for $t = T-1$, and $\forall t \in [T-2]$,

$$u_{\pi,t}(s,a) \doteq (q_{\pi,t}(s,a) - b_t(s,a))^2 + \nu_{\pi,t}(s,a) + \sum_{s'} p(s'|s,a)\mathbb{V}\left(G^b(\tau_{t+1:T-1}^{\mu_{t+1:T-1}^*}) \mid S_{t+1} = s'\right). \tag{13}$$

Notably, $u_{\pi,t}$ and $\mu_t^*$ are defined backwards and alternatively, i.e., they are defined in the order of $u_{\pi,T-1}, \mu_{T-1}^*, u_{\pi,T-2}, \mu_{T-2}^*, \ldots, u_{\pi,0}, \mu_0^*$. We now break down each term in $u_{\pi,t}(s,a)$.

1. $(q_{\pi,t}(s,a) - b_t(s,a))^2$ is the squared difference between the state-value function $q_{\pi,t}$ and the baseline function $b$. This term is always non-negative because of the square operation. Its magnitude is mainly controlled by the baseline function $b$.

2. $\nu_{\pi,t}(s,a)$ defined in (11) is the variance of the value for the next state. This term is always non-negative by the definition of variance. Its magnitude is mainly controlled by the stochasticity of the environment (i.e. transition function $p$).

3. $\sum_{s'} p(s'|s,a)\mathbb{V}\left(G^b(\tau_{t+1:T-1}^{\mu_{t+1:T-1}^*}) \mid S_{t+1} = s'\right)$ is the expected future variance given the current state $s$ and action $a$. This term is always non-negative by the definition of variance. Its magnitude is jointly controlled by the choice of behavior policy $\mu^*$, the baseline function $b$, and the transition function $p$.

$u_{\pi,t}(s,a)$ is non-negative because it is the sum of three non-negative terms. Therefore, $\sqrt{u_{\pi,t}(s,a)}$ is always well-defined. In (12), $\mu_t^*(a|s) \propto \pi_t(a|s)\sqrt{u_{\pi,t}(s,a)}$ means $\mu_t^*(a|s) \doteq \frac{\pi_t(a|s)\sqrt{u_{\pi,t}(s,a)}}{\sum_b \pi_t(b|s)\sqrt{u_{\pi,t}(s,b)}}$. If $\forall a, \pi_t(a|s)\sqrt{u_{\pi,t}(s,a)} = 0$, the denominator is zero. In this case, we use the convention to interpret $\mu_t^*(a|s)$ as a uniform distribution, i.e., $\forall a, \mu_t^*(a|s) = 1/|\mathcal{A}|$. We adopt this convention for $\propto$ in the rest of the paper to simplify the presentation. Now, we define the enlarged space $\Lambda$ as

$$\Lambda \doteq \{\mu \mid \forall t, s, a, \mu_t(a|s) = 0 \implies \pi_t(a|s)u_{\pi,t}(s,a) = 0\}. \tag{14}$$

We prove that this policy space $\Lambda$ defined above satisfies the broadness (5) and the unbiasedness (6) by the following lemmas.

**Lemma 1** (Broadness). $\forall b, \Lambda^- \subseteq \Lambda$.

Its proof is in Appendix A.1.

**Lemma 2** (Unbiasedness). $\forall b, \forall \mu \in \Lambda, \forall t, \forall s, \mathbb{E}\left[G^b(\tau_{t:T-1}^{\mu_{t:T-1}}) \mid S_t = s\right] = v_{\pi,t}(s)$.

Its proof is in Appendix A.2. After confirming the broadness and unbiasedness of the space $\Lambda$, we now prove that the behavior policy $\mu^*$ is the optimal solution for the inner optimization problem.

**Theorem 1.** *For a baseline function $b$, the behavior policy $\mu^*$ defined in (12) is an optimal solution to the optimization problems $\forall t, s$,*

$$\min_{\mu} \quad \mathbb{V}\left(G^b(\tau_{t:T-1}^{\mu_{t:T-1}}) \mid S_t = s\right)$$

$$s.t. \quad \mu \in \Lambda.$$

Its proof is in Appendix A.3. Theorem 1 proves that $\forall b$, the behavior policy $\mu^*$ (12) is the closed-form optimal solution for all $t$ and $s$. With Theorem 1, for any $t$ and $s$, we reduce the bi-level optimization problem

$$\min_{b} \min_{\mu} \quad \mathbb{V}\left(G^b(\tau_{t:T-1}^{\mu_{t:T-1}}) \mid S_t = s\right)$$

$$s.t. \quad \mu \in \Lambda$$

to a single-level unconstrained optimization problem

$$\min_{b} \quad \mathbb{V}\left(G^b(\tau_{t:T-1}^{\mu_{t:T-1}^*}) \mid S_t = s\right).$$

In this unconstrained optimization problem, we design a function $b$ that influences both the data processing estimator $G^b$ (1) and the optimal behavior policy $\mu^*$ (12). Notably, the optimal behavior policy $\mu^*$ depends on the baseline $b$ because it is tailored to a baseline function $b$ in Theorem 1. Unless otherwise noted, we omit explicitly writing this dependency in the notation of $\mu^*$ for simplicity. We show that although both $G^b$ and $\mu^*$ depend on $b$, through the mathematical proof in the appendix, the optimal baseline function $b^*$ has a concise format. Define $\forall t, s, a$,

$$b_t^*(s,a) \doteq q_{\pi,t}(s,a). \tag{15}$$

**Theorem 2.** *$b^*$ is the optimal solution to the optimization problems $\forall t, s$,*

$$\min_{b} \quad \mathbb{V}\left(G^b(\tau_{t:T-1}^{\mu_{t:T-1}^*}) \mid S_t = s\right). \tag{16}$$

Its proof is in Appendix A.4. By solving each level of the optimization problem, we show $(\mu^*, b^*)$ is the optimal solution for the bi-level optimization problem by utilizing Theorem 1 and Theorem 2.

**Theorem 3.** *$(\mu^*, b^*)$ is the optimal solution to the bi-level optimization problems $\forall t, s$,*

$$\min_{b} \min_{\mu} \quad \mathbb{V}\left(G^b(\tau_{t:T-1}^{\mu_{t:T-1}}) \mid S_t = s\right)$$

$$s.t. \quad \mu \in \Lambda.$$

*Proof.* $\forall b, \forall \mu \in \Lambda$, we have $\forall t, \forall s$

$$\mathbb{V}\left(G^b(\tau_{t:T-1}^{\mu_{t:T-1}}) \mid S_t = s\right)$$

$$\geq \mathbb{V}\left(G^b(\tau_{t:T-1}^{\mu_{t:T-1}^*}) \mid S_t = s\right) \qquad \text{(Theorem 1)}$$

$$\geq \mathbb{V}\left(G^{b^*}(\tau_{t:T-1}^{\mu_{t:T-1}^*}) \mid S_t = s\right). \qquad \text{(Theorem 2)}$$

Thus, $(\mu^*, b^*)$ achieves the minimum value of $\mathbb{V}\left(G^b(\tau_{t:T-1}^{\mu_{t:T-1}}) \mid S_t = s\right)$ for all $t$ and $s$. $\qquad\square$

## 5 VARIANCE COMPARISON

Theorem 3 shows $(\mu^*, b^*)$ is the optimal behavior policy and baseline function. This means $(\mu^*, b^*)$ is superior to any other choice of $(\mu, b)$. In this section, we further quantify its superiority. We quantify the variance reduction in reinforcement learning. We show that the variance reduction compounds over each step, bringing substantial advantages. Specifically, we provide a theoretical comparison of our method—the doubly optimal estimator—with the following baselines: (1) the on-policy Monte Carlo estimator, (2) the offline data informed estimator (Liu and Zhang, 2024), and (3) the doubly robust estimator (Jiang and Li, 2016). We use $u_t^{b^*}$ to denote $u_t$ (13) using $b^*$ as the baseline function. First, we compare our off-policy estimator with the on-policy Monte Carlo estimator (ON).

**Theorem 4.** $\forall t, s,$

$$
\mathbb{V}\left(G^{PDIS}(\tau_{t:T-1}^{\pi_{t:T-1}}) \mid S_t = s\right) - \mathbb{V}\left(G^{b^*}(\tau_{t:T-1}^{\mu_{t:T-1}^*}) \mid S_t = s\right)
$$

$$
= \underbrace{\mathbb{V}_{A_t \sim \pi_t}\left(\sqrt{u_t^{b^*}(S_t, A_t)} \mid S_t = s\right)}_{(4.1)} + \underbrace{\mathbb{V}_{A_t \sim \pi_t}\left(q_{\pi,t}(S_t, A_t) \mid S_t = s\right)}_{(4.2)} + \underbrace{\delta_t^{ON, \, ours}(s)}_{(4.3)},
$$

where $\delta_t^{ON, \, ours}(s) \doteq 0$ for $t = T - 1$ and $\forall t \in [T-2]$, $\delta_t^{ON, \, ours}(s) \doteq$

$$
\mathbb{E}_{A_t \sim \pi_t, S_{t+1}}\left[\mathbb{V}\left(G^{PDIS}(\tau_{t+1:T-1}^{\pi_{t+1:T-1}}) \mid S_{t+1}\right) - \mathbb{V}\left(G^{b^*}(\tau_{t+1:T-1}^{\mu_{t+1:T-1}^*}) \mid S_{t+1}\right) \mid S_t = s\right].
$$

*Moreover, we prove $\forall t, s, \delta_t^{ON, \, ours}(s) \geq 0$ meaning the variance reduction in future steps is compounded into the current step.*

Its proof is in Appendix A.5. In Theorem 4, we show that the variance reduction of our method includes three sources. First, a part of the future variance (4.1) is eliminated by choosing an optimal behavior policy $\mu^*$. Second, the variance of the $q$ function (4.2) is eliminated by the optimal baseline function $b^*$. Third, the variance reduction in the future step (4.3) is compounded into the current step.

Next, the following theorem quantifies the variance reduction of our method compared with the offline data informed (ODI) method in Liu and Zhang (2024). Because the behavior policy $\mu^*$ is tailored for the baseline function $b$, we use $\mu^{*,b}$ to denote $\mu^*$ with a baseline function $b$ and $\mu^{*,\text{PDIS}}$ to denote $\mu^*$ with no baseline function (i.e., the plain PDIS estimator considered in offline data informed (ODI) method (Liu and Zhang, 2024)).

**Theorem 5.** $\forall t, s,$

$$
\mathbb{V}\left(G^{PDIS}(\tau_{t:T-1}^{\mu_{t:T-1}^{*,\text{PDIS}}}) \mid S_t = s\right) - \mathbb{V}\left(G^{b^*}(\tau_{t:T-1}^{\mu_{t:T-1}^{*,b^*}}) \mid S_t = s\right)
$$

$$
\geq \underbrace{\mathbb{V}_{A_t \sim \mu_t^{*,\text{PDIS}}}\left(\rho_t q_{\pi,t}(S_t, A_t) \mid S_t\right)}_{(5.1)} + \underbrace{\delta_t^{ODI, \, ours}(s)}_{(5.2)},
$$

where $\delta_t^{ODI, \, ours}(s) \doteq 0$ for $t = T - 1$ and $\forall t \in [T-2]$, $\delta_t^{ODI, \, ours}(s) \doteq$

$$
\mathbb{E}_{A_t \sim \mu_t^{*,\text{PDIS}}, S_{t+1}}\left[\rho_t^2\left[\mathbb{V}\left(G^{PDIS}(\tau_{t+1:T-1}^{\mu_{t+1:T-1}^{*,\text{PDIS}}}) \mid S_{t+1}\right) - \mathbb{V}\left(G^{PDIS}(\tau_{t+1:T-1}^{\mu_{t+1:T-1}^{*,b^*}}) \mid S_{t+1}\right)\right] \mid S_t\right].
$$

*Moreover, we prove $\forall t, s, \delta_t^{ODI, \, ours}(s) \geq 0$ meaning the variance reduction in future steps is compounded into the current step.*

Its proof is in Appendix A.6. The variance reduction of our estimator includes two sources. First, the variance of the $q$ function (5.1) is eliminated. Second, the variance reduction in the future step (5.2) is compounded to the current step.

We also quantify the variance reduction of our estimator with the doubly robust (DR) estimator defined in Jiang and Li (2016). Since Jiang and Li (2016) does not specify any candidate behavior policy, we leverage the conventional wisdom, supposing they use the canonical target policy $\pi$ as the data-collecting policy.

**Theorem 6.** $\forall t, s,$

$$\mathbb{V}\left(G^{b^*}(\tau_{t:T-1}^{\pi_{t:T-1}}) \mid S_t = s\right) - \mathbb{V}\left(G^{b^*}(\tau_{t:T-1}^{\mu^*_{t:T-1}}) \mid S_t = s\right)$$

$$= \underbrace{\mathbb{V}_{A_t \sim \pi_t}\left(\sqrt{u_t^{b^*}(S_t, A_t)} \mid S_t = s\right)}_{(6.1)} + \underbrace{\delta_t^{DR,\ ours}(s)}_{(6.2)},$$

where $\delta_t^{DR,\ ours}(s) \doteq 0$ for $t = T - 1$ and $\forall t \in [T-2]$, $\delta^{DR,\ ours_t}(s) \doteq$

$$\mathbb{E}_{A_t \sim \pi_t, S_{t+1}}\left[\mathbb{V}\left(G^{b^*}(\tau_{t+1:T-1}^{\pi_{t+1:T-1}}) \mid S_{t+1}\right) - \mathbb{V}\left(G^{b^*}(\tau_{t+1:T-1}^{\mu^*_{t+1:T-1}}) \mid S_{t+1}\right) \mid S_t\right].$$

*Moreover, we prove $\forall t, s, \delta_t^{DR,\ ours}(s) \geq 0$ meaning the variance reduction in future steps is compounded into the current step.*

Its proof is in Appendix A.7. Similarly, there are two sources of the variance reduction for our method. First, with an optimal behavior policy $\mu^*$, we eliminate a part of the future variance (6.1). Second, the variance reduction in the future steps (6.2) is compounded to the current step.

## 6 LEARNING CLOSED-FORM BEHAVIOR POLICIES

---
**Algorithm 1:** Doubly Optimal (DOpt) Policy Evaluation

---
1: **Input:** a target policy $\pi$,
       an offline dataset $\mathcal{D} = \{(t_i, s_i, a_i, r_i, s_i')\}_{i=1}^m$
2: **Output:** a behavior policy $\mu^*$,
       a baseline function $b^*$
3: Approximate $q_{\pi,t}$ from $\mathcal{D}$ using offline RL methods (e.g. Fitted Q-Evaluation)
4: Construct $\nu_{\pi,t}$ from $\mathcal{D}$ by (38)
5: Construct $\mathcal{D}_\nu \doteq \{(t_i, s_i, a_i, \nu_i, s_i')\}_{i=1}^m$
6: Approximate $u_{\pi,t}$ from $\mathcal{D}_\nu$ by Lemma 3
7: **Return:** $\mu_t^*(a|s) \propto \pi_t(a|s)\sqrt{u_{\pi,t}(s,a)}$, $b_t^*(s,a) = q_{\pi,t}(s,a)$

---

In this section, we present an efficient Algorithm 1 to learn our doubly optimal method including the optimal behavior policy $\mu^*$ and the optimal baseline function $b^*$. Specifically, we learn $(\mu^*, b^*)$ from offline data pairs. By definition (15), we can apply any off-the-shelf offline policy evaluation methods to learn $b_t^*(s,a) \doteq q_{\pi,t}(s,a)$ (e.g., Fitted Q-Evaluation (Le et al., 2019)). By (12), $\mu_t^*(a|s) \propto \pi_t(a|s)\sqrt{u_{\pi,t}(s,a)}$, where $u$ is defined in (13) as

$$u_{\pi,t}(s,a) \doteq (q_{\pi,t}(s,a) - b_t(s,a))^2 + \nu_{\pi,t}(s,a) + \sum_{s'} p(s'|s,a)\mathbb{V}\left(G^b(\tau_{t+1:T-1}^{\mu^*_{t+1:T-1}}) \mid S_{t+1} = s'\right).$$

Learning $u$ from this perspective is very inefficient because it requires the approximation of the complicated variance term $\mathbb{V}\left(G^b(\tau_{t+1:T-1}^{\mu^*_{t+1:T-1}}) \mid S_{t+1} = s'\right)$ regarding future trajectories. To solve this problem, we propose the following recursive form of $u$.

**Lemma 3** (Recursive form of $u$). *With $b = b^*$, when $t = T - 1$, $\forall s, a$, $u_{\pi,t}(s,a) = 0$, when $t \in [T-2]$, $\forall s, a$,*

$$u_{\pi,t}(s,a) = \nu_{\pi,t}(s,a) + \sum_{s',a'} \rho_{t+1} p(s'|s,a)\pi_{t+1}(a'|s')u_{\pi,t+1}(s',a').$$

Its proof is in Appendix A.8. This lemma allows us to learn $u$ recursively without approximating the complicated trajectory variance. Subsequently, the desired optimal behavior policy $\mu^*$ can be easily calculated using (12). To ensure broad applicability, we utilize the behavior policy-agnostic offline learning setting (Nachum et al., 2019), in which the offline data consists of $\{(t_i, s_i, a_i, r_i, s_i')\}_{i=1}^m$, with $m$ previously logged data tuples. Those tuples can be generated by various unknown behavior policies, and they are not required to form a complete trajectory. In the $i$-th data tuple, $t_i$ represents

the time step, $s_i$ is the state at time step $t_i$, $a_i$ is the action executed, $r_i$ is the sampled reward, and $s_i'$ is the successor state. In this paper, we learn $(\mu^*, b^*)$ from cheaply available offline data using Fitted $Q$-Evaluation (FQE, (Le et al., 2019)), but our framework is ready to integrate any state-of-the-art offline policy evaluation technique. As for constructing $\nu$, we use the learned $q$ function and $r_i$, $s_i'$ from the data tuples, according to the derivation (38) in Appendix B.

## 7    EMPIRICAL RESULTS

In this section, we show the empirical comparison between our methods and three baselines: **(1)** the on-policy Monte Carlo estimator, **(2)** the offline data informed estimator (ODI, Liu and Zhang (2024)), and **(3)** the doubly robust estimator (DR, Jiang and Li (2016)). In the doubly robust estimator, because they do not design a tailored behavior policy, we leverage the conventional wisdom to use the target policy $\pi$ as the behavior policy. Given previously logged offline data, we learn our optimal behavior policy and the optimal baseline tuple $(\mu^*, b^*)$ using Algorithm 1. All baseline methods learn their required quantities from the same offline dataset to ensure fair comparisons. We use the behavior policy $\mu^*$ for data collection and the baseline $b^*$ for data processing. Since our method reduces variance in both the data-collecting and the data-processing phases, we name our method doubly optimal (DOpt) policy evaluation. More experiment details are in Appendix B.

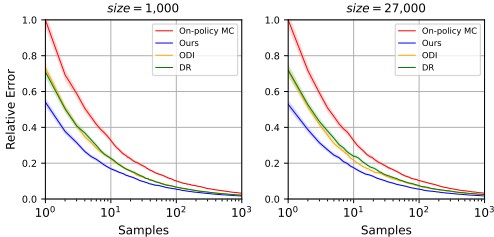

Figure 1: Results on Gridworld. Each curve is averaged over 900 runs (30 target policies, each having 30 independent runs). Shaded regions denote standard errors and are invisible for some curves because they are too small.

| Env Size | On-policy MC | Ours | ODI | DR |
|---|---|---|---|---|
| 1,000 | 1.000 | **0.274** | 0.467 | 0.450 |
| 27,000 | 1.000 | **0.283** | 0.481 | 0.541 |

Table 1: Relative variance of estimators on Gridworld. The relative variance is defined as the variance of each estimator divided by the variance of the on-policy Monte Carlo estimator. Numbers are averaged over 900 independent runs (30 target policies, each having 30 independent runs).

**Gridworld:** We begin by conducting experiments in Gridworld with $n^3$ states, i.e., an $n \times n$ grid with $n$ as the time horizon. The number of states in this Gridworld environment scales cubically with $n$, offering a suitable tool to test algorithm scalability. We choose Gridworld with $n^3 = 1,000$ and $n^3 = 27,000$, which are the largest Gridworld environment tested among related works (Jiang and Li, 2016; Hanna et al., 2017; Liu and Zhang, 2024). We use randomly generated reward functions with 30 randomly generated target policies. The offline data is generated by various unknown policies to simulate cheaply available but segmented offline data. Because MC methods use each episode as one empirical return sample, we view each episode as one online sample. We report the *relative error* of the four methods against the number of online samples. This relative error is the estimation error normalized by the estimation error of the on-policy Monte Carlo estimator after the first episode. Thus, the relative error of the on-policy Monte Carlo estimator starts from 1.

Figure 1 shows our method outperforms all baselines by a large margin. The blue line in the graph is below all other lines, indicating that our method requires fewer samples to achieve the same accuracy. This is because our designed $(\mu^*, b^*)$ substantially reduces estimation variance. In Table 1, we quantify such variance reduction, showing our method reduces variance by around 75% in both Gridworld with size 1,000 and 27,000.

One observation is that DR performs slightly better than ODI in smaller Gridworld but is slightly worse in larger Gridworld, which shows that there might be no dominating relationship between those two methods. Meanwhile, our method is superior to both approaches because the variance reduction of our method comes from both data-collecting and data-processing.

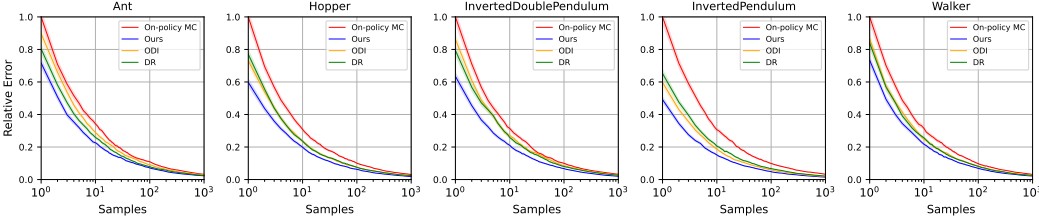

Figure 2: Results on MuJoCo. Each curve is averaged over 900 independent runs (30 target policies, each having 30 independent runs). Shaded regions denote standard errors and are invisible for some curves because they are too small.

|  | On-policy MC | **Ours** | ODI | DR | Saved Episodes Percentage |
|---|---|---|---|---|---|
| Ant | 1000 | **492** | 810 | 636 | (1000 - 492)/1000 = **50.8%** |
| Hopper | 1000 | **372** | 544 | 582 | (1000 - 372)/1000 = **62.8%** |
| I. D. Pendulum | 1000 | **426** | 727 | 651 | (1000 - 426)/1000 = **57.4%** |
| I. Pendulum | 1000 | **225** | 356 | 439 | (1000 - 225)/1000 = **77.5%** |
| Walker | 1000 | **475** | 705 | 658 | (1000 - 475)/1000 = **52.5%** |

Table 2: Episodes needed to achieve the same of estimation accuracy that on-policy Monte Carlo achieves with 1000 episodes. Standard errors are plotted in Figure 2. Each number is averaged over 900 independet runs.

**MuJoCo:** We also conduct experiments in MuJoCo robot simulation tasks (Todorov et al., 2012). MuJoCo is a physics engine containing various stochastic environments, where the goal is to control a robot to achieve different behaviors such as walking, jumping, and balancing. Figure 2 shows our method is consistently better than all baselines in various MuJoCo robot environments. Table 2 shows our method requires substantially fewer samples to achieve the same estimation accuracy compared with the on-policy Monte Carlo method. Specifically, our method saves 50.8% to 77.5% of online interactions in different tasks, achieving state-of-the-art performance in policy evaluation.

It is worth mentioning that our method is robust to hyperparameter choices—all hyperparameters required to learn $(\mu^*, b^*)$ in our method are tuned offline and stay the same across all environments.

## 8  CONCLUSION

Due to the sequential nature of reinforcement learning, policy evaluation often suffers from large variance and requires massive data to achieve the desired level of accuracy. In this work, we design an optimal combination of data-collecting policy $\mu^*$ and data-processing baseline $b^*$.

Theoretically, we prove our method considers larger policy space (Lemma 1), and is unbiased (Lemma 2) and optimal (Theorem 3). Further, we mathematically quantify the superiority of our method in variance reduction compared with existing methods (Theorem 4, 5, 6).

Empirically, compared with previous best-performing methods, we show our method reduces variance substantially in a broad range of environments, achieving state-of-the-art performance in policy evaluation.

One limitation is, as there is no free lunch, if the offline data size is too small—perhaps consisting of just a single data tuple—the behavior policy and baseline approximated by our method may be inaccurate. In this case, we recommend on-policy evaluation. The future work of our paper is to extend the variance reduction technique to temporal difference learning.

ACKNOWLEDGEMENTS

This work is supported in part by the US National Science Foundation (NSF) under grants III-2128019 and SLES-2331904.

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

## A PROOFS

### A.1 PROOF OF LEMMA 1

*Proof.* Given any baseline function $b$, $\forall \mu \in \Lambda^-$, $\forall t, s, a$,

$$\mu_t(a|s) = 0$$
$$\implies \pi_t(a|s) = 0 \qquad \qquad \text{(Definition of } \Lambda^-)$$
$$\implies \pi_t(a|s)u_{\pi,t}(s,a) = 0.$$

This shows $\mu \in \Lambda$. Thus, $\Lambda^- \subseteq \Lambda$. $\qquad \square$

### A.2 PROOF OF LEMMA 2

To prove Lemma 2, we first prove the following auxiliary lemma.

**Lemma 4.** $\forall b, \forall \mu \in \Lambda, \forall t, s$,

$$\mathbb{E}_{A_t \sim \mu_t(\cdot|S_t)} \left[ \rho_t[q_{\pi,t}(S_t, A_t) - b_t(S_t, A_t)] + \bar{b}_t(S_t) \mid S_t = s \right] = \mathbb{E}_{A_t \sim \pi_t(\cdot|S_t)} \left[ q_{\pi,t}(S_t, A_t) \mid S_t = s \right].$$

*Proof.* We fix any baseline function $b$. Because $\mu \in \Lambda$, $\forall t, s, a$,

$$\mu_t(a|s) = 0$$
$$\implies \pi_t(a|s)u_{\pi,t}(s,a) = 0$$
$$\implies \pi_t(a|s)\left[ (q_{\pi,t}(s,a) - b_t(s,a))^2 + \nu_{\pi,t}(s,a) + \sum_{s'} p(s'|s,a)\mathbb{V}\left(G^b(\tau_{t+1:T-1}^{\mu_{t+1:T-1}^*}) \mid S_{t+1} = s'\right) \right] = 0 \tag{By (13)}$$

$$\implies \pi_t(a|s)(q_{\pi,t}(s,a) - b_t(s,a))^2 = 0$$
$$\qquad (\nu_{\pi,t}(s,a) \text{ and } \sum_{s'} p(s'|s,a)\mathbb{V}\left(G^b(\tau_{t+1:T-1}^{\mu_{t+1:T-1}^*}) \mid S_{t+1} = s'\right) \text{ are non-negative})$$
$$\implies \pi_t(a|s)(q_{\pi,t}(s,a) - b_t(s,a)) = 0. \tag{17}$$

Then, we have

$$\mathbb{E}_{A_t \sim \mu_t(\cdot|S_t)} \left[ \rho_t[q_{\pi,t}(S_t, A_t) - b_t(S_t, A_t)] + \bar{b}_t(S_t) \mid S_t = s \right]$$

$$= \mathbb{E}_{A_t \sim \mu_t(\cdot|S_t)} \left[ \frac{\pi_t(A_t|S_t)}{\mu_t(A_t|S_t)}[q_{\pi,t}(S_t, A_t) - b_t(S_t, A_t)] + \bar{b}_t(S_t) \mid S_t = s \right]$$

$$= \sum_{a \in \{a|\mu_t(a|s)>0\}} \mu_t(a|s)\left[ \frac{\pi_t(a|s)}{\mu_t(a|s)}[q_{\pi,t}(s,a) - b_t(s,a)] + \bar{b}_t(s) \right]$$

$$= \sum_{a \in \{a|\mu_t(a|s)>0\}} \pi_t(a|s)[q_{\pi,t}(s,a) - b_t(s,a)] + \sum_{a \in \{a|\mu_t(a|s)>0\}} \mu_t(a|s)\bar{b}_t(s)$$

$$= \sum_{a \in \{a|\mu_t(a|s)>0\}} \pi_t(a|s)[q_{\pi,t}(s,a) - b_t(s,a)] + \bar{b}_t(s) \sum_{a \in \{a|\mu_t(a|s)>0\}} \mu_t(a|s)$$

$$= \sum_{a \in \{a|\mu_t(a|s)>0\}} \pi_t(a|s)[q_{\pi,t}(s,a) - b_t(s,a)] + \bar{b}_t(s)$$

$$= \sum_{a \in \{a|\mu_t(a|s)>0\}} \pi_t(a|s)[q_{\pi,t}(s,a) - b_t(s,a)]$$

$$\qquad + \sum_{a \in \{a|\mu_t(a|s)=0\}} \pi_t(a|s)[q_{\pi,t}(s,a) - b_t(s,a)] + \bar{b}_t(s) \tag{By (17)}$$

$$= \sum_a \pi_t(a|s)[q_{\pi,t}(s,a) - b_t(s,a)] + \bar{b}_t(s)$$

$$= \sum_a \pi_t(a|s)q_{\pi,t}(s,a) - \bar{b}_t(s) + \bar{b}_t(s) \tag{By (2)}$$

$$=\mathbb{E}_{A\sim\pi}[q_{\pi,t}(S_t, A_t) \mid S_t = s].$$

$\square$

Now, we are ready to prove Lemma 2.

**Lemma 2** (Unbiasedness). $\forall b, \forall \mu \in \Lambda, \forall t, \forall s, \mathbb{E}\left[G^b(\tau_{t:T-1}^{\mu_t:T-1}) \mid S_t = s\right] = v_{\pi,t}(s).$

*Proof.* Fix any baseline function $b$. We proceed via induction.

For $t = T - 1, \forall \mu \in \Lambda, \forall s$, we have

$$\begin{aligned}
&\mathbb{E}\left[G^b(\tau_{t:T-1}^{\mu_t:T-1}) \mid S_t = s\right] \\
=&\mathbb{E}_{A_t\sim\mu_t(\cdot|S_t)}\left[\rho_t[R_{t+1} - b_t(S_t, A_t)] + \bar{b}_t(S_t) \mid S_t\right] \\
=&\mathbb{E}_{A_t\sim\mu_t(\cdot|S_t)}\left[\rho_t[q_{\pi,t}(S_t, A_t) - b_t(S_t, A_t)] + \bar{b}_t(S_t) \mid S_t\right] \\
=&\mathbb{E}_{A_t\sim\pi_t(\cdot|S_t)}\left[q_{\pi,t}(S_t, A_t) \mid S_t\right] & \text{(Lemma 4)} \\
=&v_{\pi,t}(S_t).
\end{aligned}$$

For $t \in [T - 2]$, assuming that Lemma 2 holds for $t + 1$, we have $\forall \mu \in \Lambda, \forall s$,

$$\mathbb{E}\left[G^b(\tau_{t+1:T-1}^{\mu_{t+1}:T-1}) \mid S_{t+1} = s\right] = v_{\pi,t+1}(s).$$

Then, $\forall t$,

$$\begin{aligned}
&\mathbb{E}\left[G^b(\tau_{t:T-1}^{\mu_t:T-1}) \mid S_t = s\right] \\
=&\mathbb{E}\left[\rho_t\left(R_{t+1} + G^b(\tau_{t+1:T-1}^{\mu_{t+1}:T-1}) - b_t(S_t, A_t)\right) + \bar{b}_t(S_t)) \mid S_t\right] & \text{(By (1))} \\
=&\mathbb{E}\left[\rho_t(R_{t+1} - b_t(S_t, A_t)) + \bar{b}_t(S_t) \mid S_t\right] + \mathbb{E}\left[\rho_t G^b(\tau_{t+1:T-1}^{\mu_{t+1}:T-1}) \mid S_t\right] \\
=&\mathbb{E}\left[\rho_t(R_{t+1} - b_t(S_t, A_t)) + \bar{b}_t(S_t) \mid S_t\right] \\
&+ \mathbb{E}_{A_t\sim\mu_t(\cdot|S_t),S_{t+1}\sim p(\cdot|S_t,A_t)}\left[\mathbb{E}\left[\rho_t G^b(\tau_{t+1:T-1}^{\mu_{t+1}:T-1}) \mid S_t, A_t, S_{t+1}\right] \mid S_t\right] \\
&\hspace{6cm} \text{(Law of Iterated Expectation)} \\
=&\mathbb{E}\left[\rho_t(R_{t+1} - b_t(S_t, A_t)) + \bar{b}_t(S_t) \mid S_t\right] \\
&+ \mathbb{E}_{A_t\sim\mu_t(\cdot|S_t),S_{t+1}\sim p(\cdot|S_t,A_t)}\left[\rho_t\mathbb{E}\left[G^b(\tau_{t+1:T-1}^{\mu_{t+1}:T-1}) \mid S_{t+1}\right] \mid S_t\right] \\
&\hspace{4cm} \text{(Conditional independence and Markov property)} \\
=&\mathbb{E}\left[\rho_t(R_{t+1} - b_t(S_t, A_t)) + \bar{b}_t(S_t) \mid S_t\right] + \mathbb{E}_{A_t\sim\mu_t(\cdot|S_t),S_{t+1}\sim p(\cdot|S_t,A_t)}\left[\rho_t v_{\pi,t+1}(S_{t+1}) \mid S_t\right] \\
&\hspace{8cm} \text{(Inductive hypothesis)} \\
=&\mathbb{E}_{A_t\sim\mu_t(\cdot|S_t)}\left[\rho_t[q_{\pi,t}(S_t, A_t) - b_t(S_t, A_t)] + \bar{b}_t(S_t) \mid S_t\right] & \text{(Definition of } q_{\pi,t}) \\
=&\mathbb{E}_{A_t\sim\pi_t(\cdot|S_t)}\left[q_{\pi,t}(S_t, A_t) \mid S_t\right] & \text{(Lemma 4)} \\
=&v_{\pi,t}(s),
\end{aligned}$$

which completes the proof. $\square$

### A.3 PROOF OF THEOREM 1

To prove Theorem 1, we first characterize the variance of the off-policy estimator in a recursive form.

**Lemma 5.** $\forall b, \forall \mu \in \Lambda$, *for* $t = T - 1$,

$$\mathbb{V}\left(G^b(\tau_{t:T-1}^{\mu_t:T-1}) \mid S_t\right) = \mathbb{E}_{A_t\sim\mu_t}\left[\rho_t^2[q_{\pi,t}(S_t, A_t) - b_t(S_t, A_t)]^2 \mid S_t\right] - [v_{\pi,t}(S_t) - \bar{b}_t(S_t)]^2;$$

*For* $t \in [T - 2]$,

$$\begin{aligned}
&\mathbb{V}\left(G^b(\tau_{t:T-1}^{\mu_t:T-1}) \mid S_t\right) \\
=&\mathbb{E}_{A_t\sim\mu_t}\left[\rho_t^2\left(\mathbb{E}_{S_{t+1}}\left[\mathbb{V}\left(G^b(\tau_{t+1:T-1}^{\mu_{t+1}:T-1}) \mid S_{t+1}\right) \mid S_t, A_t\right] + \nu_t(S_t, A_t) + [q_{\pi,t}(S_t, A_t) - b_t(S_t, A_t)]^2\right) \mid S_t\right] \\
&- [v_{\pi,t}(S_t) - \bar{b}_t(S_t)]^2.
\end{aligned}$$

*Proof.* When $t \in [T-2]$, we have

$$\mathbb{V}\left(G^b(\tau_{t:T-1}^{\mu_{t:T-1}}) \mid S_t\right)$$
$$=\mathbb{E}_{A_t}\left[\mathbb{V}\left(G^b(\tau_{t:T-1}^{\mu_{t:T-1}}) \mid S_t, A_t\right) \mid S_t\right] + \mathbb{V}_{A_t}\left(\mathbb{E}\left[G^b(\tau_{t:T-1}^{\mu_{t:T-1}}) \mid S_t, A_t\right] \mid S_t\right)$$
$$\text{(Law of total variance)}$$
$$=\mathbb{E}_{A_t}\left[\mathbb{V}\left(\rho_t\left[r(S_t, A_t) + G^b(\tau_{t+1:T-1}^{\mu_{t+1:T-1}}) - b_t(S_t, A_t)\right] + \bar{b}_t(S_t) \mid S_t, A_t\right) \mid S_t\right]$$
$$+ \mathbb{V}_{A_t}\left(\mathbb{E}\left[\rho_t\left[r(S_t, A_t) + G^b(\tau_{t+1:T-1}^{\mu_{t+1:T-1}}) - b_t(S_t, A_t)\right] + \bar{b}_t(S_t)) \mid S_t, A_t\right] \mid S_t\right) \quad \text{(By (1))}$$
$$=\mathbb{E}_{A_t}\left[\rho_t^2 \mathbb{V}\left(G^b(\tau_{t+1:T-1}^{\mu_{t+1:T-1}}) \mid S_t, A_t\right) \mid S_t\right]$$
$$+ \mathbb{V}_{A_t}\left(\rho_t[r(S_t, A_t) + \mathbb{E}\left[G^b(\tau_{t+1:T-1}^{\mu_{t+1:T-1}}) \mid S_t, A_t\right] - b_t(S_t, A_t)] + \bar{b}_t(S_t) \mid S_t\right)$$
$$\quad (r(S_t, A_t), b_t(S_t, A_t), \bar{b}_t(S_t) \text{ are constant given } S_t, A_t)$$
$$=\mathbb{E}_{A_t}\left[\rho_t^2 \mathbb{V}\left(G^b(\tau_{t+1:T-1}^{\mu_{t+1:T-1}}) \mid S_t, A_t\right) \mid S_t\right]$$
$$+ \mathbb{V}_{A_t}\left(\rho_t[r(S_t, A_t) + \mathbb{E}\left[v_{\pi,t+1}(S_{t+1}) \mid S_t, A_t\right] - b_t(S_t, A_t)] + \bar{b}_t(S_t) \mid S_t\right) \quad \text{(Lemma 2)}$$
$$=\mathbb{E}_{A_t}\left[\rho_t^2 \mathbb{V}\left(G^b(\tau_{t+1:T-1}^{\mu_{t+1:T-1}}) \mid S_t, A_t\right) \mid S_t\right]$$
$$+ \mathbb{V}_{A_t}\left(\rho_t[q_{\pi,t}(S_t, A_t) - b_t(S_t, A_t)] + \bar{b}_t(S_t) \mid S_t\right). \quad \text{(Defintion of } q_{\pi,t}) \quad (18)$$

For the inner part of the first term, we have

$$\mathbb{V}\left(G^b(\tau_{t+1:T-1}^{\mu_{t+1:T-1}}) \mid S_t, A_t\right)$$
$$=\mathbb{E}_{S_{t+1}}\left[\mathbb{V}\left(G^b(\tau_{t+1:T-1}^{\mu_{t+1:T-1}}) \mid S_t, A_t, S_{t+1}\right) \mid S_t, A_t\right]$$
$$+ \mathbb{V}_{S_{t+1}}\left(\mathbb{E}\left[G^b(\tau_{t+1:T-1}^{\mu_{t+1:T-1}}) \mid S_t, A_t, S_{t+1}\right] \mid S_t, A_t\right) \quad \text{(Law of total variance)}$$
$$=\mathbb{E}_{S_{t+1}}\left[\mathbb{V}\left(G^b(\tau_{t+1:T-1}^{\mu_{t+1:T-1}}) \mid S_{t+1}\right) \mid S_t, A_t\right] + \mathbb{V}_{S_{t+1}}\left(\mathbb{E}\left[G^b(\tau_{t+1:T-1}^{\mu_{t+1:T-1}}) \mid S_{t+1}\right] \mid S_t, A_t\right)$$
$$\text{(Markov property)}$$
$$=\mathbb{E}_{S_{t+1}}\left[\mathbb{V}\left(G^b(\tau_{t+1:T-1}^{\mu_{t+1:T-1}}) \mid S_{t+1}\right) \mid S_t, A_t\right] + \mathbb{V}_{S_{t+1}}\left(v_{\pi,t+1}(S_{t+1}) \mid S_t, A_t\right) \quad \text{(Lemma 2)}$$
$$=\mathbb{E}_{S_{t+1}}\left[\mathbb{V}\left(G^b(\tau_{t+1:T-1}^{\mu_{t+1:T-1}}) \mid S_{t+1}\right) \mid S_t, A_t\right] + \nu_t(S_t, A_t). \quad \text{(By (11))} \quad (19)$$

For the second term, we have

$$\nu_t(S_t, A_t)$$
$$=\mathbb{V}_{A_t}\left(\rho_t[q_{\pi,t}(S_t, A_t) - b_t(S_t, A_t)] + \bar{b}_t(S_t) \mid S_t\right) \quad \text{(By (11))}$$
$$=\mathbb{E}_{A_t}\left[\left(\rho_t[q_{\pi,t}(S_t, A_t) - b_t(S_t, A_t)] + \bar{b}_t(S_t)\right)^2 \mid S_t\right]$$
$$- \left(\mathbb{E}_{A_t}\left[\rho_t[q_{\pi,t}(S_t, A_t) - b_t(S_t, A_t)] + \bar{b}_t(S_t) \mid S_t\right]\right)^2$$
$$=\mathbb{E}_{A_t}\left[\left(\rho_t[q_{\pi,t}(S_t, A_t) - b_t(S_t, A_t)] + \bar{b}_t(S_t)\right)^2 \mid S_t\right] - v_{\pi,t}(S_t)^2. \quad \text{(Lemma 4)}$$
$$=\mathbb{E}_{A_t}\left[\rho_t^2[q_{\pi,t}(S_t, A_t) - b_t(S_t, A_t)]^2 \mid S_t\right] + 2\bar{b}_t(S_t)\mathbb{E}_{A_t}\left[\rho_t[q_{\pi,t}(S_t, A_t) - b_t(S_t, A_t)] \mid S_t\right]$$
$$+ \bar{b}_t(S_t)^2 - v_{\pi,t}(S_t)^2$$
$$=\mathbb{E}_{A_t}\left[\rho_t^2[q_{\pi,t}(S_t, A_t) - b_t(S_t, A_t)]^2 \mid S_t\right] + 2\bar{b}_t(S_t)\mathbb{E}_{A_t}\left[\rho_t[q_{\pi,t}(S_t, A_t) - b_t(S_t, A_t)] + \bar{b}_t(S_t) \mid S_t\right]$$
$$- 2\bar{b}_t(S_t)^2 + \bar{b}_t(S_t)^2 - v_{\pi,t}(S_t)^2$$
$$=\mathbb{E}_{A_t}\left[\rho_t^2[q_{\pi,t}(S_t, A_t) - b_t(S_t, A_t)]^2 \mid S_t\right] + 2\bar{b}_t(S_t)v_{\pi,t}(S_t)$$
$$- \bar{b}_t(S_t)^2 - v_{\pi,t}(S_t)^2 \quad \text{(Lemma 2)}$$
$$=\mathbb{E}_{A_t}\left[\rho_t^2[q_{\pi,t}(S_t, A_t) - b_t(S_t, A_t)]^2 \mid S_t\right] - (v_{\pi,t}(S_t) - \bar{b}_t(S_t))^2. \quad (20)$$

Plugging (19) and (20) back to (18) gives

$$\mathbb{V}\left(G^b(\tau_{t:T-1}^{\mu_{t:T-1}}) \mid S_t\right)$$
$$=\mathbb{E}_{A_t}\left[\rho_t^2 \mathbb{V}\left(G^b(\tau_{t+1:T-1}^{\mu_{t+1:T-1}}) \mid S_t, A_t\right) \mid S_t\right]$$
$$+ \mathbb{V}_{A_t}\left(\rho_t[q_{\pi,t}(S_t, A_t) - b_t(S_t, A_t)] + \bar{b}_t(S_t) \mid S_t\right) \quad \text{(By (18))}$$

$$
\begin{aligned}
=& \mathbb{E}_{A_t} \left[ \rho_t^2 \left( \mathbb{E}_{S_{t+1}} \left[ \mathbb{V} \left( G^b(\tau_{t+1:T-1}^{\mu_{t+1:T-1}}) \mid S_{t+1} \right) \mid S_t, A_t \right] + \nu_t(S_t, A_t) \right) \mid S_t \right] \\
& + \mathbb{V}_{A_t} \left( \rho_t[q_{\pi,t}(S_t, A_t) - b_t(S_t, A_t)] + \bar{b}_t(S_t) \mid S_t \right) \quad \text{(By (19))} \\
=& \mathbb{E}_{A_t} \left[ \rho_t^2 \left( \mathbb{E}_{S_{t+1}} \left[ \mathbb{V} \left( G^b(\tau_{t+1:T-1}^{\mu_{t+1:T-1}}) \mid S_{t+1} \right) \mid S_t, A_t \right] + \nu_t(S_t, A_t) \right) \mid S_t \right] \\
& + \mathbb{E}_{A_t} \left[ \rho_t^2[q_{\pi,t}(S_t, A_t) - b_t(S_t, A_t)]^2 \mid S_t \right] - (v_{\pi,t}(S_t) - \bar{b}_t(S_t))^2 \quad \text{(By (20))} \\
=& \mathbb{E}_{A_t} \left[ \rho_t^2 \left( \mathbb{E}_{S_{t+1}} \left[ \mathbb{V} \left( G^b(\tau_{t+1:T-1}^{\mu_{t+1:T-1}^*}) \mid S_{t+1} \right) \mid S_t, A_t \right] + \nu_t(S_t, A_t) + [q_{\pi,t}(S_t, A_t) - b_t(S_t, A_t)]^2 \right) \mid S_t \right]
\end{aligned}
$$

$$
- [v_{\pi,t}(S_t) - \bar{b}_t(S_t)]^2.
$$

When $t = T - 1$, we have

$$
\begin{aligned}
& \mathbb{V} \left( G^b(\tau_{t:T-1}^{\mu_{t:T-1}}) \mid S_t \right) \\
=& \mathbb{V} \left( \rho_t[r(S_t, A_t) - b_t(S_t, A_t)] + \bar{b}_t(S_t) \mid S_t \right) \quad \text{(By (1))} \\
=& \mathbb{V} \left( \rho_t[q_{\pi,t}(S_t, A_t) - b_t(S_t, A_t)] + \bar{b}_t(S_t) \mid S_t \right) \\
=& \mathbb{E}_{A_t} \left[ \rho_t^2[q_{\pi,t}(S_t, A_t) - b_t(S_t, A_t)]^2 \mid S_t \right] - (v_{\pi,t}(S_t) - \bar{b}_t(S_t))^2,
\end{aligned}
$$

which completes the proof. $\qquad\square$

We now restate Theorem 1 and give its proof.

**Theorem 1.** *For a baseline function $b$, the behavior policy $\mu^*$ defined in (12) is an optimal solution to the optimization problems $\forall t, s$,*

$$
\min_{\mu} \quad \mathbb{V} \left( G^b(\tau_{t:T-1}^{\mu_{t:T-1}}) \mid S_t = s \right)
$$

$$
s.t. \quad \mu \in \Lambda.
$$

*Proof.* Fix a baseline function $b$. $\forall t, s, a$,

$$
\begin{aligned}
& \mu_t^*(a|s) = 0 \\
& \implies \pi_t(a|s) \sqrt{u_{\pi,t}(s, a)} = 0 \quad \text{(By (12))} \\
& \implies \pi_t(a|s) u_{\pi,t}(s, a) = 0.
\end{aligned}
$$

Thus, $\mu^* \in \Lambda$.

$\forall t, \forall \mu \in \Lambda$, we have an unbiasedness on $\sqrt{u_{\pi,t}(s, a)}$.

$$
\begin{aligned}
& \mathbb{E}_{A_t \sim \mu_t} \left[ \rho_t \sqrt{u_{\pi,t}(S_t, A_t)} \mid S_t = s \right] \\
=& \sum_{a \in \{a | \mu_t(a|s) > 0\}} \mu_t(a|s) \frac{\pi_t(a|s)}{\mu_t(a|s)} \sqrt{u_{\pi,t}(s, a)} \\
=& \sum_{a \in \{a | \mu_t(a|s) > 0\}} \pi_t(a|s) \sqrt{u_{\pi,t}(s, a)} \\
=& \sum_{a \in \{a | \mu_t(a|s) > 0\}} \pi_t(a|s) \sqrt{u_{\pi,t}(s, a)} + \sum_{a \in \{a | \mu_t(a|s) = 0\}} \pi_t(a|s) \sqrt{u_{\pi,t}(s, a)} \\
& \quad (\forall \mu \in \Lambda, \mu_t(a|s) = 0 \implies \pi_t(a|s) u_{\pi,t}(s, a) = 0 \text{ by (14)}) \\
=& \mathbb{E}_{A_t \sim \pi_t} \left[ \sqrt{u_{\pi,t}(S_t, A_t)} \mid S_t = s \right]. \quad (21)
\end{aligned}
$$

We prove the optimality of the behavior policy $\mu^*$ via induction.

When $t = T - 1$, $\forall \mu \in \Lambda$, $\forall s$, the variance of the off-policy estimator has the following lower bound

$$
\mathbb{V} \left( G^b(\tau_{t:T-1}^{\mu_{t:T-1}}) \mid S_t = s \right)
$$

$$=\mathbb{E}_{A_t\sim\mu_t}\left[\rho_t^2[q_{\pi,t}(S_t,A_t)-b_t(S_t,A_t)]^2\mid S_t\right]-(v_{\pi,t}(S_t)-\bar{b}_t(S_t))^2 \qquad\text{(Lemma 5)}$$

$$=\mathbb{E}_{A_t\sim\mu_t}\left[\rho_t^2 u_{\pi,t}(S_t,A_t)\mid S_t\right]-(v_{\pi,t}(S_t)-\bar{b}_t(S_t))^2 \qquad\text{(By (13))}$$

$$\geq\mathbb{E}_{A_t\sim\mu_t}\left[\rho_t\sqrt{u_{\pi,t}(S_t,A_t)}\mid S_t\right]^2-(v_{\pi,t}(S_t)-\bar{b}_t(S_t))^2 \qquad\text{(By Jensen's Inequality)}$$

$$=\mathbb{E}_{A_t\sim\pi_t}\left[\sqrt{u_{\pi,t}(S_t,A_t)}\mid S_t\right]^2-(v_{\pi,t}(S_t)-\bar{b}_t(S_t))^2. \qquad\text{(By (21))}$$

For any state $s$, the variance of the off-policy estimator with the behavior policy $\mu^*$ achieves this lower bound by the following derivations.

$$\mathbb{V}\left(G^b(\tau_{t:T-1}^{\mu_{t:T-1}^*})\mid S_t=s\right) \qquad(22)$$

$$=\mathbb{E}_{A_t\sim\mu_t^*}\left[\rho_t^2[q_{\pi,t}(S_t,A_t)-b_t(S_t,A_t)]^2\mid S_t\right]-(v_{\pi,t}(S_t)-\bar{b}_t(S_t))^2 \qquad\text{(Lemma 5)}$$

$$=\mathbb{E}_{A_t\sim\mu_t^*}\left[\rho_t^2 u_{\pi,t}(S_t,A_t)\mid S_t\right]-(v_{\pi,t}(S_t)-\bar{b}_t(S_t))^2. \qquad\text{(By (13))}$$

For the first term, we have

$$\mathbb{E}_{A_t\sim\mu_t^*}\left[\rho_t^2 u_{\pi,t}(S_t,A_t)\mid S_t\right]$$

$$=\sum_a \frac{\pi_t(a|S_t)^2}{\mu_t^*(a|S_t)}u_{\pi,t}(S_t,a)$$

$$=\sum_a \pi_t(a|S_t)\sqrt{u_{\pi,t}(S_t,a)}\sum_b \pi_t(S_t,b)\sqrt{u_{\pi,t}(S_t,b)} \qquad\text{(By (12))}$$

$$=\mathbb{E}_{A_t\sim\pi_t}\left[\sqrt{u_{\pi,t}(S_t,A_t)}\mid S_t\right]^2. \qquad(23)$$

Plugging (23) back to (22), we obtain

$$\mathbb{V}\left(G^b(\tau_{t:T-1}^{\mu_{t:T-1}^*})\mid S_t=s\right)$$

$$=\mathbb{E}_{A_t\sim\mu_t^*}\left[\rho_t^2 u_{\pi,t}(S_t,A_t)\mid S_t\right]-(v_{\pi,t}(S_t)-\bar{b}_t(S_t))^2$$

$$=\mathbb{E}_{A_t\sim\pi_t}\left[\sqrt{u_{\pi,t}(S_t,A_t)}\mid S_t\right]^2-(v_{\pi,t}(S_t)-\bar{b}_t(S_t))^2.$$

Thus, the behavior policy $\mu^*$ defined in (12) is an optimal solution to the optimization problems

$$\min_\mu \quad \mathbb{V}\left(G^b(\tau_{t:T-1}^{\mu_{t:T-1}})\mid S_t=s\right)$$

$$\text{s.t.}\quad \mu\in\Lambda$$

for $t=T-1$ and all $s$.

When $t\in[T-2]$, we proceed via induction. The inductive hypothesis is that the behavior policy $\mu^*$ is an optimal solution to the optimization problems

$$\min_\mu \quad \mathbb{V}\left(G^b(\tau_{t+1:T-1}^{\mu_{t+1:T-1}})\mid S_t=s\right)$$

$$\text{s.t.}\quad \mu\in\Lambda$$

for all $s$.

To complete the induction, we prove that the behavior policy $\mu^*$ is an optimal solution to the optimization problems

$$\min_\mu \quad \mathbb{V}\left(G^b(\tau_{t:T-1}^{\mu_{t:T-1}})\mid S_t=s\right)$$

$$\text{s.t.}\quad \mu\in\Lambda$$

for all $s$.

$\forall \mu \in \Lambda$, $\forall s$, the variance of the off-policy estimator has the following lower bound

$$\mathbb{V}\left(G^b(\tau_{t:T-1}^{\mu_{t:T-1}}) \mid S_t = s\right)$$
$$= \mathbb{E}_{A_t \sim \mu_t}\left[\rho_t^2\left(\mathbb{E}_{S_{t+1}}\left[\mathbb{V}\left(G^b(\tau_{t+1:T-1}^{\mu_{t+1:T-1}}) \mid S_{t+1}\right) \mid S_t, A_t\right] + \nu_t(S_t, A_t) + [q_{\pi,t}(S_t, A_t) - b_t(S_t, A_t)]^2\right) \mid S_t\right]$$

$$- [v_{\pi,t}(S_t) - \bar{b}_t(S_t)]^2 \qquad\qquad (\text{Lemma } 5)$$
$$\geq \mathbb{E}_{A_t \sim \mu_t}\left[\rho_t^2\left(\mathbb{E}_{S_{t+1}}\left[\mathbb{V}\left(G^b(\tau_{t+1:T-1}^{\mu_{t+1:T-1}^*}) \mid S_{t+1}\right) \mid S_t, A_t\right] + \nu_t(S_t, A_t) + [q_{\pi,t}(S_t, A_t) - b_t(S_t, A_t)]^2\right) \mid S_t\right]$$

$$- [v_{\pi,t}(S_t) - \bar{b}_t(S_t)]^2 \qquad\qquad (\text{Indutive Hypothesis})$$
$$= \mathbb{E}_{A_t \sim \mu_t}\left[\rho_t^2 u_{\pi,t}(S_t, A_t) \mid S_t\right] - (v_{\pi,t}(S_t) - \bar{b}_t(S_t))^2 \qquad\qquad (\text{By } (13))$$
$$\geq \mathbb{E}_{A_t \sim \mu_t}\left[\rho_t \sqrt{u_{\pi,t}(S_t, A_t)} \mid S_t\right]^2 - (v_{\pi,t}(S_t) - \bar{b}_t(S_t))^2 \qquad\qquad (\text{By Jensen's Inequality})$$
$$= \mathbb{E}_{A_t \sim \pi_t}\left[\sqrt{u_{\pi,t}(S_t, A_t)} \mid S_t\right]^2 - (v_{\pi,t}(S_t) - \bar{b}_t(S_t))^2. \qquad\qquad (\text{By } (21))$$

For any state $s$, the variance of the off-policy estimator with the behavior policy $\mu^*$ achieves the lower bound by the following derivations.

$$\mathbb{V}\left(G^b(\tau_{t:T-1}^{\mu_{t:T-1}^*}) \mid S_t = s\right)$$
$$= \mathbb{E}_{A_t \sim \mu_t}\left[\rho_t^2\left(\mathbb{E}_{S_{t+1}}\left[\mathbb{V}\left(G^b(\tau_{t+1:T-1}^{\mu_{t+1:T-1}^*}) \mid S_{t+1}\right) \mid S_t, A_t\right] + \nu_t(S_t, A_t) + [q_{\pi,t}(S_t, A_t) - b_t(S_t, A_t)]^2\right) \mid S_t\right]$$

$$- [v_{\pi,t}(S_t) - \bar{b}_t(S_t)]^2 \qquad\qquad (\text{Lemma } 5)$$
$$= \mathbb{E}_{A_t \sim \mu_t^*}\left[\rho_t^2 u_{\pi,t}(S_t, A_t) \mid S_t\right] - (v_{\pi,t}(S_t) - \bar{b}_t(S_t))^2 \qquad\qquad (\text{By } (13))$$
$$= \mathbb{E}_{A_t \sim \pi_t}\left[\sqrt{u_{\pi,t}(S_t, A_t)} \mid S_t\right]^2 - (v_{\pi,t}(S_t) - \bar{b}_t(S_t))^2. \qquad\qquad (\text{By } (23))$$

Thus, the behavior policy $\mu^*$ defined in (12) is an optimal solution to the optimization problems

$$\min_{\mu} \quad \mathbb{V}\left(G^b(\tau_{t:T-1}^{\mu_{t:T-1}}) \mid S_t = s\right)$$
$$\text{s.t.} \quad \mu \in \Lambda$$

for $t$ and all $s$.

This completes the induction. $\qquad\qquad\square$

## A.4 PROOF OF THEOREM 2

In this proof, to differentiate different $\mu^*$ with different baseline functions $b$, we use $\mu^{*,b}$ to denote the corresponding $\mu^*$ when using a function $b$ as the baseline function. $G^b$, $u_{\pi,t}^b$, and $\Lambda^b$ are defined following the same convention. We first present an auxiliary lemma.

**Lemma 6.** $\forall b$, $\forall \mu \in \Lambda^b$, $\forall t$,

$$\mathbb{E}_{A_t \sim \mu_t}\left[\rho_t^2[q_{\pi,t}(S_t, A_t) - b_t(S_t, A_t)]^2 \mid S_t\right] - [v_{\pi,t}(S_t) - \bar{b}_t(S_t)]^2$$
$$= \mathbb{V}_{A_t \sim \mu_t}\left(\rho_t[q_{\pi,t}(S_t, A_t) - b_t(S_t, A_t)] \mid S_t\right) \qquad\qquad (24)$$

*Proof.* $\forall b$, $\forall \mu \in \Lambda^b$, $\forall t$,

$$\mathbb{E}_{A_t \sim \mu_t}\left[\rho_t^2[q_{\pi,t}(S_t, A_t) - b_t(S_t, A_t)]^2 \mid S_t\right] - [v_{\pi,t}(S_t) - \bar{b}_t(S_t)]^2$$
$$= \mathbb{V}_{A_t \sim \mu_t}\left(\rho_t[q_{\pi,t}(S_t, A_t) - b_t(S_t, A_t)] \mid S_t\right) + \mathbb{E}_{A_t \sim \mu_t}\left[\rho_t[q_{\pi,t}(S_t, A_t) - b_t(S_t, A_t)] \mid S_t\right]^2$$
$$- [v_{\pi,t}(S_t) - \bar{b}_t(S_t)]^2$$
$$= \mathbb{V}_{A_t \sim \mu_t}\left(\rho_t[q_{\pi,t}(S_t, A_t) - b_t(S_t, A_t)] \mid S_t\right) + \mathbb{E}_{A_t \sim \mu_t}\left[\rho_t[q_{\pi,t}(S_t, A_t) - b_t(S_t, A_t)] \mid S_t\right]^2$$

$$- \left[ \mathbb{E}_{A_t \sim \mu_t(\cdot|S_t)} \left[ \rho_t[q_{\pi,t}(S_t, A_t) - b_t(S_t, A_t)] + \bar{b}_t(S_t) \mid S_t \right] - \bar{b}_t(S_t) \right]^2$$

$$\text{(Definition of } q_{\pi,t}, \text{ Lemma 4)}$$

$$= \mathbb{V}_{A_t \sim \mu_t} \left( \rho_t[q_{\pi,t}(S_t, A_t) - b_t(S_t, A_t)] \mid S_t \right) + \mathbb{E}_{A_t \sim \mu_t} \left[ \rho_t[q_{\pi,t}(S_t, A_t) - b_t(S_t, A_t)] \mid S_t \right]^2$$

$$- \mathbb{E}_{A_t \sim \mu_t(\cdot|S_t)} \left[ \rho_t[q_{\pi,t}(S_t, A_t) - b_t(S_t, A_t)] \mid S_t \right]^2$$

$$= \mathbb{V}_{A_t \sim \mu_t} \left( \rho_t[q_{\pi,t}(S_t, A_t) - b_t(S_t, A_t)] \mid S_t \right).$$

$$\square$$

We now restate Theorem 2 and give its proof.

**Theorem 2.** $b^*$ *is the optimal solution to the optimization problems* $\forall t, s,$

$$\min_b \quad \mathbb{V}\left( G^b(\tau_{t:T-1}^{\mu_{t:T-1}^*}) \mid S_t = s \right). \tag{16}$$

*Proof.* We prove this by induction on the time step $t$.

When $t = T - 1$, $\forall s$, the optimization problem (16) has the following lower bound

$$\mathbb{V}(G^b(\tau_{t:T-1}^{\mu_{t:T-1}^{*,b}}) \mid S_0 = s)$$

$$= \mathbb{E}_{A_t \sim \mu_t^{*,b}} \left[ \rho_t^2[q_{\pi,t}(S_t, A_t) - b_t(S_t, A_t)]^2 \mid S_t \right] - [v_{\pi,t}(S_t) - \bar{b}_t(S_t)]^2 \quad \text{(Lemma 5)}$$

$$= \mathbb{V}_{A_t \sim \mu_t^{*,b}} \left( \rho_t[q_{\pi,t}(S_t, A_t) - b_t(S_t, A_t)] \mid S_t \right) \quad \text{(Lemma 6)}$$

$$\geq 0. \quad \text{(Variance non-negativity)}$$

When using $b^*$ as the baseline, we achieve this lower bound.

$$\mathbb{V}(G^{b^*}(\tau_{t:T-1}^{\mu_{t:T-1}^{*,b^*}}) \mid S_0 = s)$$

$$= \mathbb{E}_{A_t \sim \mu_t^{*,b^*}} \left[ \rho_t^2[q_{\pi,t}(S_t, A_t) - b_t^*(S_t, A_t)]^2 \mid S_t \right] - [v_{\pi,t}(S_t) - \bar{b}_t^*(S_t)]^2 \quad \text{(Lemma 5)}$$

$$= \mathbb{V}_{A_t \sim \mu_t^{*,b^*}} \left( \rho_t[q_{\pi,t}(S_t, A_t) - b_t^*(S_t, A_t)] \mid S_t \right) \quad \text{(Lemma 6)}$$

$$= \mathbb{V}_{A_t \sim \mu_t^{*,b^*}} \left( \rho_t[q_{\pi,t}(S_t, A_t) - q_{\pi,t}(S_t, A_t)] \mid S_t \right) \quad \text{(Definition of } b^* \text{ (15))}$$

$$= 0.$$

When $t \in [T - 2]$, we proceed via induction. The inductive hypothesis is that the baseline function $b^*$ is an optimal solution to the optimization problems

$$\min_b \quad \mathbb{V}\left( G^b(\tau_{t+1:T-1}^{\mu_{t+1:T-1}^{*,b}}) \mid S_t = s \right)$$

for all $s$. Notice that we have

$$\Lambda^b \subseteq \Lambda^{b^*}. \tag{25}$$

This is because $\forall s, a$,

$$u_{\pi,t}^b(s, a)$$

$$= (q_{\pi,t}(s, a) - b_t(s, a))^2 + \nu_{\pi,t}(s, a) + \sum_{s'} p(s'|s, a) \mathbb{V}\left( G^b(\tau_{t+1:T-1}^{\mu_{t+1:T-1}^{*,b}}) \mid S_{t+1} = s' \right) \quad \text{(By (13))}$$

$$\geq \nu_{\pi,t}(s, a) + \sum_{s'} p(s'|s, a) \mathbb{V}\left( G^b(\tau_{t+1:T-1}^{\mu_{t+1:T-1}^{*,b}}) \mid S_{t+1} = s' \right)$$

$$\geq \nu_{\pi,t}(s, a) + \sum_{s'} p(s'|s, a) \mathbb{V}\left( G^{b^*}(\tau_{t+1:T-1}^{\mu_{t+1:T-1}^{*,b^*}}) \mid S_{t+1} = s' \right) \quad \text{(Inductive Hypothesis)}$$

$$\geq u_{\pi,t}^{b^*}(s, a).$$

Thus, $\forall \mu \in \Lambda^b$, we have $\forall s, a$

$$\mu(a|s) = 0$$

$$\implies \pi(a|s)u_{\pi,t}^b(s,a) = 0$$
$$\implies \pi(a|s)u_{\pi,t}^{b^*}(s,a) = 0.$$

This shows

$$\Lambda^b \subseteq \Lambda^{b^*}.$$

$\forall b$, the optimization problem (16) has the following lower bound

$$\mathbb{V}\left( G^b(\tau_{t:T-1}^{\mu_{t:T-1}^{*,b}}) \mid S_0 = s \right)$$

$$=\mathbb{E}_{A_t \sim \mu_t^{*,b}} \left[ \rho_t^2 \left( \mathbb{E}_{S_{t+1}} \left[ \mathbb{V}\left( G^b(\tau_{t+1:T-1}^{\mu_{t+1:T-1}^{*,b}}) \mid S_{t+1} \right) \mid S_t, A_t \right] + \nu_t(S_t, A_t) \right) \mid S_t \right]$$
$$\quad + \mathbb{E}_{A_t \sim \mu_t^{*,b}} \left[ \rho_t^2[q_{\pi,t}(S_t, A_t) - b_t(S_t, A_t)]^2 \mid S_t \right] - [v_{\pi,t}(S_t) - \bar{b}_t(S_t)]^2 \qquad \text{(Lemma 5)}$$

$$\geq \mathbb{E}_{A_t \sim \mu_t^{*,b}} \left[ \rho_t^2 \left( \mathbb{E}_{S_{t+1}} \left[ \mathbb{V}\left( G^{b^*}(\tau_{t+1:T-1}^{\mu_{t+1:T-1}^{*,b^*}}) \mid S_{t+1} \right) \mid S_t, A_t \right] + \nu_t(S_t, A_t) \right) \mid S_t \right]$$
$$\quad + \mathbb{E}_{A_t \sim \mu_t^{*,b}} \left[ \rho_t^2[q_{\pi,t}(S_t, A_t) - b_t(S_t, A_t)]^2 \mid S_t \right] - [v_{\pi,t}(S_t) - \bar{b}_t(S_t)]^2$$
$$\qquad\qquad \text{(Inductive hypothesis)}$$

$$=\mathbb{E}_{A_t \sim \mu_t^{*,b}} \left[ \rho_t^2 \left( \mathbb{E}_{S_{t+1}} \left[ \mathbb{V}\left( G^{b^*}(\tau_{t+1:T-1}^{\mu_{t+1:T-1}^{*,b^*}}) \mid S_{t+1} \right) \mid S_t, A_t \right] + \nu_t(S_t, A_t) \right) \mid S_t \right]$$
$$\quad + \mathbb{V}_{A_t \sim \mu_t^{*,b}} \left( \rho_t[q_{\pi,t}(S_t, A_t) - b_t(S_t, A_t)] \mid S_t \right) \qquad \text{(Lemma 6)}$$

$$\geq \mathbb{E}_{A_t \sim \mu_t^{*,b}} \left[ \rho_t^2 \left( \mathbb{E}_{S_{t+1}} \left[ \mathbb{V}\left( G^{b^*}(\tau_{t+1:T-1}^{\mu_{t+1:T-1}^{*,b^*}}) \mid S_{t+1} \right) \mid S_t, A_t \right] + \nu_t(S_t, A_t) \right) \mid S_t \right]$$
$$\qquad\qquad \text{(Variance non-negativity)}$$

$$=\mathbb{E}_{A_t \sim \mu_t^{*,b}} \left[ \rho_t^2 u_{\pi,t}^{b^*}(S_t, A_t) \mid S_t \right] \qquad \text{(By (13))}$$

$$\geq \mathbb{E}_{A_t \sim \mu_t^{*,b}} \left[ \rho_t \sqrt{u_{\pi,t}^{b^*}(S_t, A_t)} \mid S_t \right]^2 \qquad \text{(Jensen's inequality)}$$

$$=\mathbb{E}_{A_t \sim \pi_t} \left[ \sqrt{u_{\pi,t}^{b^*}(S_t, A_t)} \mid S_t \right]^2 \qquad \text{(By (21) and (25))}$$

When setting $\forall s, \forall a, b_t^*(s,a) \doteq q_{\pi,t}(s,a)$ as the baseline, we achieve this lower bound.

$$\mathbb{V}\left( G^{b^*}(\tau_{t:T-1}^{\mu_{t:T-1}^{*,b^*}}) \mid S_0 = s \right)$$

$$=\mathbb{E}_{A_t \sim \mu_t^{*,b^*}} \left[ \rho_t^2 \left( \mathbb{E}_{S_{t+1}} \left[ \mathbb{V}\left( G^{b^*}(\tau_{t+1:T-1}^{\mu_{t+1:T-1}^{*,b^*}}) \mid S_{t+1} \right) \mid S_t, A_t \right] + \nu_t(S_t, A_t) \right) \mid S_t \right]$$
$$\quad + \mathbb{E}_{A_t \sim \mu_t^{*,b^*}} \left[ \rho_t^2[q_{\pi,t}(S_t, A_t) - b_t^*(S_t, A_t)]^2 \mid S_t \right] - [v_{\pi,t}(S_t) - \bar{b}_t^*(S_t)]^2 \qquad \text{(Lemma 5)}$$

$$=\mathbb{E}_{A_t \sim \mu_t^{*,b^*}} \left[ \rho_t^2 \left( \mathbb{E}_{S_{t+1}} \left[ \mathbb{V}\left( G^{b^*}(\tau_{t+1:T-1}^{\mu_{t+1:T-1}^{*,b^*}}) \mid S_{t+1} \right) \mid S_t, A_t \right] + \nu_t(S_t, A_t) \right) \mid S_t \right]$$
$$\quad + \mathbb{V}_{A_t \sim \mu_t^{*,b^*}} \left( \rho_t[q_{\pi,t}(S_t, A_t) - b_t^*(S_t, A_t)] \mid S_t \right) \qquad \text{(Lemma 6)}$$

$$=\mathbb{E}_{A_t \sim \mu_t^{*,b^*}} \left[ \rho_t^2 \left( \mathbb{E}_{S_{t+1}} \left[ \mathbb{V}\left( G^{b^*}(\tau_{t+1:T-1}^{\mu_{t+1:T-1}^{*,b^*}}) \mid S_{t+1} \right) \mid S_t, A_t \right] + \nu_t(S_t, A_t) \right) \mid S_t \right]$$
$$\quad + \mathbb{V}_{A_t \sim \mu_t^{*,b^*}} \left( \rho_t[q_{\pi,t}(S_t, A_t) - q_{\pi,t}(S_t, A_t)] \mid S_t \right) \qquad \text{(Definition of } b^* \text{ (15))}$$

$$=\mathbb{E}_{A_t \sim \mu_t^{*,b^*}} \left[ \rho_t^2 \left( \mathbb{E}_{S_{t+1}} \left[ \mathbb{V}\left( G^{b^*}(\tau_{t+1:T-1}^{\mu_{t+1:T-1}^{*,b^*}}) \mid S_{t+1} \right) \mid S_t, A_t \right] + \nu_t(S_t, A_t) \right) \mid S_t \right]$$

$$=\mathbb{E}_{A_t \sim \mu_t^{*,b^*}} \left[ \rho_t^2 u_{\pi,t}^{b^*}(S_t, A_t) \mid S_t \right] \qquad \text{(By (13))}$$

$$=\mathbb{E}_{A_t \sim \pi_t} \left[ \sqrt{u_{\pi,t}^{b^*}(S_t, A_t)} \mid S_t \right]^2 \qquad \text{(By (12))}$$

Thus, $b^*$ is the optimal solution to the optimization problem

$$\min_b \quad \mathbb{V}\left(G^b(\tau_{t:T-1}^{\mu_{t:T-1}^*}) \mid S_t = s\right)$$

for all $t$ and $s$.

$\square$

### A.5 PROOF OF THEOREM 4

*Proof.* Use $u_t^{b^*}$ to denote $u_t$ (13) using $b^*$ as the baseline function. Then, by (13), for $t = T - 1$,

$$u_t^{b^*}(s,a) = [q_{\pi,t}(s,a) - b_t(s,a)]^2 = 0. \tag{26}$$

For $t \in [T-2]$,

$$u_t^{b^*}(s,a)$$
$$=(q_{\pi,t}(s,a) - b_t(s,a))^2 + \nu_{\pi,t}(s,a) + \sum_{s'} p(s'|s,a)\mathbb{V}\left(G^{b^*}(\tau_{t+1:T-1}^{\mu_{t+1:T-1}^{*,b^*}}) \mid S_{t+1} = s'\right)$$
$$\text{(By (13))}$$
$$=\nu_{\pi,t}(s,a) + \sum_{s'} p(s'|s,a)\mathbb{V}\left(G^{b^*}(\tau_{t+1:T-1}^{\mu_{t+1:T-1}^{*,b^*}}) \mid S_{t+1} = s'\right). \tag{By (15)} \tag{27}$$

The variance of $G^{b^*}(\tau_{t:T-1}^{\mu_{t:T-1}^*})$ has $\forall s$, for $t = T - 1$,

$$\mathbb{V}\left(G^{b^*}(\tau_{t:T-1}^{\mu_{t:T-1}^*}) \mid S_t = s\right) \tag{28}$$
$$=\mathbb{E}_{A_t \sim \mu_t^*}\left[\rho_t^2[q_{\pi,t}(S_t, A_t) - b_t^*(S_t, A_t)]^2 \mid S_t\right] - [v_{\pi,t}(S_t) - \bar{b}^*{}_t(S_t)]^2 \tag{Lemma 5}$$
$$=0 \tag{Definition of $b^*$ (15)}$$
$$=\mathbb{E}_{A_t \sim \mu_t^*}\left[\rho_t^2 u_t^{b^*}(S_t, A_t) \mid S_t\right] \tag{By (26)}$$
$$=\mathbb{E}_{A_t \sim \pi_t}\left[\sqrt{u_t^{b^*}(S_t, A_t)} \mid S_t\right]^2. \tag{By (23)}$$

For $t \in [T-2]$,

$$\mathbb{V}\left(G^{b^*}(\tau_{t:T-1}^{\mu_{t:T-1}^*}) \mid S_t = s\right) \tag{29}$$
$$=\mathbb{E}_{A_t \sim \mu_t^*}\left[\rho_t^2\left(\mathbb{E}_{S_{t+1}}\left[\mathbb{V}\left(G^{b^*}(\tau_{t+1:T-1}^{\mu_{t+1:T-1}^*}) \mid S_{t+1}\right) \mid S_t, A_t\right] + \nu_t(S_t, A_t) + [q_{\pi,t}(S_t, A_t) - b_t(S_t, A_t)]^2\right) \mid S_t\right]$$
$$\text{(Lemma 5)}$$

$$- [v_{\pi,t}(S_t) - \bar{b}_t(S_t)]^2$$
$$=\mathbb{E}_{A_t \sim \mu_t^*}\left[\rho_t^2\left(\mathbb{E}_{S_{t+1}}\left[\mathbb{V}\left(G^{b^*}(\tau_{t+1:T-1}^{\mu_{t+1:T-1}^*}) \mid S_{t+1}\right) \mid S_t, A_t\right] + \nu_t(S_t, A_t)\right) \mid S_t\right]$$
$$+ \mathbb{V}_{A_t \sim \mu_t^*}\left(\rho_t[q_{\pi,t}(S_t, A_t) - b_t^*(S_t, A_t)] \mid S_t\right) \tag{Lemma 6}$$
$$=\mathbb{E}_{A_t \sim \mu_t^*}\left[\rho_t^2\left(\mathbb{E}_{S_{t+1}}\left[\mathbb{V}\left(G^{b^*}(\tau_{t+1:T-1}^{\mu_{t+1:T-1}^*}) \mid S_{t+1}\right) \mid S_t, A_t\right] + \nu_t(S_t, A_t)\right) \mid S_t\right] \tag{By (15)}$$
$$=\mathbb{E}_{A_t \sim \mu_t^*}\left[\rho_t^2 u_t^{b^*}(S_t, A_t) \mid S_t\right] \tag{By (27)}$$
$$=\mathbb{E}_{A_t \sim \pi_t}\left[\sqrt{u_t^{b^*}(S_t, A_t)} \mid S_t\right]^2. \tag{By (23)}$$

The variance of $G^{\text{PDIS}}(\tau_{t:T-1}^{\pi_{t:T-1}})$ has $\forall s$, for $t - T - 1$,

$$\mathbb{V}\left(G^{\text{PDIS}}(\tau_{t:T-1}^{\pi_{t:T-1}}) \mid S_t = s\right) \tag{30}$$
$$=\mathbb{E}_{A_t \sim \pi_t}\left[q_{\pi,t}(S_t, A_t)^2 \mid S_t\right] - v_{\pi,t}(S_t)^2 \tag{Lemma 5 with $b = 0$ and on-policy}$$

$$
=\mathbb{V}_{A_t\sim\pi_t}\left(q_{\pi,t}(S_t,A_t)\mid S_t\right) \tag{Lemma 6 with $b=0$ and on-policy}
$$

$$
=\mathbb{E}_{A_t\sim\pi_t}\left[u_t^{b^*}(S_t,A_t)\mid S_t\right]+\mathbb{V}_{A_t\sim\pi_t}\left(q_{\pi,t}(S_t,A_t)\mid S_t\right). \tag{By (27)}
$$

For $t\in[T-2]$,

$$
\mathbb{V}\left(G^{\mathrm{PDIS}}(\tau_{t:T-1}^{\pi_{t:T-1}})\mid S_t=s\right) \tag{31}
$$

$$
=\mathbb{E}_{A_t\sim\pi_t}\left[\mathbb{E}_{S_{t+1}}\left[\mathbb{V}\left(G^{\mathrm{PDIS}}(\tau_{t+1:T-1}^{\pi_{t+1:T-1}})\mid S_{t+1}\right)\mid S_t,A_t\right]+\nu_t(S_t,A_t)+q_{\pi,t}(S_t,A_t)^2\mid S_t\right]
$$
$$
-v_{\pi,t}(S_t)^2 \tag{Lemma 5 with $b=0$}
$$

$$
=\mathbb{E}_{A_t\sim\pi_t}\left[\mathbb{E}_{S_{t+1}}\left[\mathbb{V}\left(G^{\mathrm{PDIS}}(\tau_{t+1:T-1}^{\pi_{t+1:T-1}})\mid S_{t+1}\right)\mid S_t,A_t\right]+\nu_t(S_t,A_t)\mid S_t\right]
$$
$$
+\mathbb{V}_{A_t\sim\pi_t}\left(q_{\pi,t}(S_t,A_t)\mid S_t\right) \tag{Lemma 6}
$$

$$
=\mathbb{E}_{A_t\sim\pi_t}\left[\mathbb{E}_{S_{t+1}}\left[\mathbb{V}\left(G^{b^*}(\tau_{t+1:T-1}^{\mu_{t+1:T-1}^{*,b^*}})\mid S_{t+1}\right)\mid S_t,A_t\right]+\nu_t(S_t,A_t)\mid S_t\right]
$$
$$
+\mathbb{V}_{A_t\sim\pi_t}\left(q_{\pi,t}(S_t,A_t)\mid S_t\right)
$$
$$
+\mathbb{E}_{A_t\sim\pi_t}\left[\mathbb{E}_{S_{t+1}}\left[\mathbb{V}\left(G^{\mathrm{PDIS}}(\tau_{t+1:T-1}^{\pi_{t+1:T-1}})\mid S_{t+1}\right)-\mathbb{V}\left(G^{b^*}(\tau_{t+1:T-1}^{\mu_{t+1:T-1}^{*,b^*}})\mid S_{t+1}\right)\mid S_t,A_t\right]\mid S_t\right]
$$

$$
=\mathbb{E}_{A_t\sim\pi_t}\left[u_t^{b^*}(S_t,A_t)\mid S_t\right]
$$
$$
+\mathbb{V}_{A_t\sim\pi_t}\left(q_{\pi,t}(S_t,A_t)\mid S_t\right)
$$
$$
+\mathbb{E}_{A_t\sim\pi_t}\left[\mathbb{E}_{S_{t+1}}\left[\mathbb{V}\left(G^{\mathrm{PDIS}}(\tau_{t+1:T-1}^{\pi_{t+1:T-1}})\mid S_{t+1}\right)-\mathbb{V}\left(G^{b^*}(\tau_{t+1:T-1}^{\mu_{t+1:T-1}^{*,b^*}})\mid S_{t+1}\right)\mid S_t,A_t\right]\mid S_t\right]. \tag{By (27)}
$$

Thus, for $t=T-1$, their difference is

$$
\mathbb{V}\left(G^{\mathrm{PDIS}}(\tau_{t:T-1}^{\pi_{t:T-1}})\mid S_t=s\right)-\mathbb{V}\left(G^{b^*}(\tau_{t:T-1}^{\mu_{t:T-1}^*})\mid S_t=s\right)
$$

$$
=\mathbb{E}_{A_t\sim\pi_t}\left[u_t^{b^*}(S_t,A_t)\mid S_t\right]-\mathbb{E}_{A_t\sim\pi_t}\left[\sqrt{u_t^{b^*}(S_t,A_t)}\mid S_t\right]^2
$$
$$
+\mathbb{V}_{A_t\sim\pi_t}\left(q_{\pi,t}(S_t,A_t)\mid S_t\right) \tag{By (28) and (30)}
$$
$$
=\mathbb{V}_{A_t\sim\pi_t}\left(\sqrt{u_t^{b^*}(S_t,A_t)}\mid S_t\right)+\mathbb{V}_{A_t\sim\pi_t}\left(q_{\pi,t}(S_t,A_t)\mid S_t\right).
$$

For $t\in[T-2]$,

$$
\mathbb{V}\left(G^{\mathrm{PDIS}}(\tau_{t:T-1}^{\pi_{t:T-1}})\mid S_t=s\right)-\mathbb{V}\left(G^{b^*}(\tau_{t:T-1}^{\mu_{t:T-1}^*})\mid S_t=s\right)
$$

$$
=\mathbb{E}_{A_t\sim\pi_t}\left[u_t^{b^*}(S_t,A_t)\mid S_t\right]-\mathbb{E}_{A_t\sim\pi_t}\left[\sqrt{u_t^{b^*}(S_t,A_t)}\mid S_t\right]^2
$$
$$
+\mathbb{V}_{A_t\sim\pi_t}\left(q_{\pi,t}(S_t,A_t)\mid S_t\right)
$$
$$
+\mathbb{E}_{A_t\sim\pi_t}\left[\mathbb{E}_{S_{t+1}}\left[\mathbb{V}\left(G^{\mathrm{PDIS}}(\tau_{t+1:T-1}^{\pi_{t+1:T-1}})\mid S_{t+1}\right)-\mathbb{V}\left(G^{b^*}(\tau_{t+1:T-1}^{\mu_{t+1:T-1}^*})\mid S_{t+1}\right)\mid S_t,A_t\right]\mid S_t\right]
$$
$$
\tag{By (29) and (31)}
$$

$$
=\mathbb{V}_{A_t\sim\pi_t}\left(\sqrt{u_t^{b^*}(S_t,A_t)}\mid S_t\right)
$$
$$
+\mathbb{V}_{A_t\sim\pi_t}\left(q_{\pi,t}(S_t,A_t)\mid S_t\right)
$$
$$
+\mathbb{E}_{A_t\sim\pi_t}\left[\mathbb{E}_{S_{t+1}}\left[\mathbb{V}\left(G^{\mathrm{PDIS}}(\tau_{t+1:T-1}^{\pi_{t+1:T-1}})\mid S_{t+1}\right)-\mathbb{V}\left(G^{b^*}(\tau_{t+1:T-1}^{\mu_{t+1:T-1}^*})\mid S_{t+1}\right)\mid S_t,A_t\right]\mid S_t\right].
$$

We use induction to prove $\forall t, s, \delta_t^{\text{ON, ours}}(s) \geq 0$. For $t = T - 1$,

$$\delta_t^{\text{ON, ours}}(s) = 0 \geq 0.$$

For $t \in [T - 2]$, the induction hypothesis is $\forall s$,

$$\delta_{t+1}^{\text{ON, ours}}(s) \geq 0.$$

This implies $\forall s$,

$$\mathbb{V}\left(G^{\text{PDIS}}(\tau_{t+1:T-1}^{\pi_{t+1:T-1}}) \mid S_{t+1} = s\right) - \mathbb{V}\left(G^{b^*}(\tau_{t+1:T-1}^{\mu_{t+1:T-1}^*}) \mid S_{t+1} = s\right)$$

$$=\mathbb{V}_{A_{t+1} \sim \pi_{t+1}}\left(\sqrt{u_{t+1}^{b^*}(S_{t+1}, A_{t+1})} \mid S_{t+1} = s\right)$$

$$+ \mathbb{V}_{A_{t+1} \sim \pi_{t+1}}\left(q_{\pi,t+1}(S_{t+1}, A_{t+1}) \mid S_{t+1} = s\right) + \delta_{t+1}^{\text{ON, ours}}(s)$$

$$\geq 0. \tag{32}$$

Thus, $\forall s$,

$$\delta_t^{\text{ON, ours}}(s)$$

$$=\mathbb{E}_{A_t \sim \pi_t, S_{t+1}}\left[\mathbb{V}\left(G^{\text{PDIS}}(\tau_{t+1:T-1}^{\pi_{t+1:T-1}}) \mid S_{t+1}\right) - \mathbb{V}\left(G^{b^*}(\tau_{t+1:T-1}^{\mu_{t+1:T-1}^*}) \mid S_{t+1}\right) \mid S_t = s\right]$$

$$\geq 0. \tag{by (32) \quad (33)}$$

Thus, $\forall t, s, \delta_t^{\text{ON, ours}}(s) \geq 0$.

$\square$

### A.6 PROOF OF THEOREM 5

*Proof.* The variance of $G^{\text{PDIS}}(\tau_{t:T-1}^{\mu_{t:T-1}^{*,\text{PDIS}}})$ has $\forall s$, for $t = T - 1$,

$$\mathbb{V}\left(G^{\text{PDIS}}(\tau_{t:T-1}^{\mu_{t:T-1}^{*,\text{PDIS}}}) \mid S_t = s\right) \tag{34}$$

$$=\mathbb{E}_{A_t \sim \mu_t^{*,\text{PDIS}}}\left[\rho_t^2 q_{\pi,t}(S_t, A_t)^2 \mid S_t\right] - v_{\pi,t}(S_t)^2 \tag{Lemma 5 and $b = 0$}$$

$$=\mathbb{V}_{A_t \sim \mu_t^{*,\text{PDIS}}}\left(\rho_t q_{\pi,t}(S_t, A_t) \mid S_t\right) \tag{By (24)}$$

$$=\mathbb{E}_{A_t \sim \mu_t^{*,\text{PDIS}}}\left[\rho_t^2 u_t^{b^*}(S_t, A_t) \mid S_t\right] + \mathbb{V}_{A_t \sim \mu_t^{*,\text{PDIS}}}\left(\rho_t q_{\pi,t}(S_t, A_t) \mid S_t\right) \tag{By (27)}$$

$$=\mathbb{E}_{A_t \sim \pi_t}\left[\sqrt{u_t^{b^*}(S_t, A_t)} \mid S_t\right]^2 + \mathbb{V}_{A_t \sim \mu_t^{*,\text{PDIS}}}\left(\rho_t q_{\pi,t}(S_t, A_t) \mid S_t\right). \tag{By (23)}$$

For $t \in [T - 2]$,

$$\mathbb{V}\left(G^{\text{PDIS}}(\tau_{t:T-1}^{\mu_{t:T-1}^{*,\text{PDIS}}}) \mid S_t = s\right) \tag{35}$$

$$=\mathbb{E}_{A_t \sim \mu_t^{*,\text{PDIS}}}\left[\rho_t^2\left(\mathbb{E}_{S_{t+1}}\left[\mathbb{V}\left(G^{\text{PDIS}}(\tau_{t+1:T-1}^{\mu_{t+1:T-1}^{*,\text{PDIS}}}) \mid S_{t+1}\right) \mid S_t, A_t\right] + \nu_t(S_t, A_t) + q_{\pi,t}(S_t, A_t)^2\right) \mid S_t\right]$$

$$- v_{\pi,t}(S_t)^2 \tag{Lemma 5 with $b = 0$}$$

$$=\mathbb{E}_{A_t \sim \mu_t^{*,\text{PDIS}}}\left[\rho_t^2\left(\mathbb{E}_{S_{t+1}}\left[\mathbb{V}\left(G^{\text{PDIS}}(\tau_{t+1:T-1}^{\mu_{t+1:T-1}^{*,\text{PDIS}}}) \mid S_{t+1}\right) \mid S_t, A_t\right] + \nu_t(S_t, A_t)\right) \mid S_t\right]$$

$$+ \mathbb{V}_{A_t \sim \mu_t^{*,\text{PDIS}}}\left(\rho_t q_{\pi,t}(S_t, A_t) \mid S_t\right) \tag{Lemma 6 with $b = 0$}$$

$$=\mathbb{E}_{A_t \sim \mu_t^{*,\text{PDIS}}}\left[\rho_t^2\left(\mathbb{E}_{S_{t+1}}\left[\mathbb{V}\left(G^{\text{PDIS}}(\tau_{t+1:T-1}^{\mu_{t+1:T-1}^{*,b^*}}) \mid S_{t+1}\right) \mid S_t, A_t\right] + \nu_t(S_t, A_t)\right) \mid S_t\right]$$

$$+ \mathbb{V}_{A_t \sim \mu_t^{*,\text{PDIS}}}\left(\rho_t q_{\pi,t}(S_t, A_t) \mid S_t\right)$$

$$+ \mathbb{E}_{A_t \sim \mu_t^{*,\text{PDIS}}} \left[ \rho_t^2 \left( \mathbb{E}_{S_{t+1}} \left[ \mathbb{V} \left( G^{\text{PDIS}}(\tau_{t+1:T-1}^{\mu_{t+1:T-1}^{*,\text{PDIS}}}) \mid S_{t+1} \right) - \mathbb{V} \left( G^{\text{PDIS}}(\tau_{t+1:T-1}^{\mu_{t+1:T-1}^{*,b^*}}) \mid S_{t+1} \right) \mid S_t, A_t \right] \right) \mid S_t \right]$$

$$\geq \mathbb{E}_{A_t \sim \mu_t^{*,b^*}} \left[ \rho_t^2 \left( \mathbb{E}_{S_{t+1}} \left[ \mathbb{V} \left( G^{\text{PDIS}}(\tau_{t+1:T-1}^{\mu_{t+1:T-1}^{*,b^*}}) \mid S_{t+1} \right) \mid S_t, A_t \right] + \nu_t(S_t, A_t) \right) \mid S_t \right]$$
$$+ \mathbb{V}_{A_t \sim \mu_t^{*,\text{PDIS}}} \left( \rho_t q_{\pi,t}(S_t, A_t) \mid S_t \right)$$
$$+ \mathbb{E}_{A_t \sim \mu_t^{*,\text{PDIS}}} \left[ \rho_t^2 \left( \mathbb{E}_{S_{t+1}} \left[ \mathbb{V} \left( G^{\text{PDIS}}(\tau_{t+1:T-1}^{\mu_{t+1:T-1}^{*,\text{PDIS}}}) \mid S_{t+1} \right) - \mathbb{V} \left( G^{\text{PDIS}}(\tau_{t+1:T-1}^{\mu_{t+1:T-1}^{*,b^*}}) \mid S_{t+1} \right) \mid S_t, A_t \right] \right) \mid S_t \right]$$
$$(\forall \mu_t^{*,\text{PDIS}} \in \Lambda_t, \mu_t^{*,b^*} \text{ achieves the minimum value of the first term in } \Lambda_t)$$

$$= \mathbb{E}_{A_t \sim \mu_t^{*,b^*}} \left[ \rho_t^2 u_t^{b^*}(S_t, A_t) \mid S_t \right]$$
$$+ \mathbb{V}_{A_t \sim \mu_t^{*,\text{PDIS}}} \left( \rho_t q_{\pi,t}(S_t, A_t) \mid S_t \right)$$
$$+ \mathbb{E}_{A_t \sim \mu_t^{*,\text{PDIS}}} \left[ \rho_t^2 \left( \mathbb{E}_{S_{t+1}} \left[ \mathbb{V} \left( G^{\text{PDIS}}(\tau_{t+1:T-1}^{\mu_{t+1:T-1}^{*,\text{PDIS}}}) \mid S_{t+1} \right) - \mathbb{V} \left( G^{\text{PDIS}}(\tau_{t+1:T-1}^{\mu_{t+1:T-1}^{*,b^*}}) \mid S_{t+1} \right) \mid S_t, A_t \right] \right) \mid S_t \right]$$
$$\text{(By (27))}$$

$$= \mathbb{E}_{A_t \sim \pi_t} \left[ \sqrt{u_t^{b^*}(S_t, A_t)} \mid S_t \right]^2$$
$$+ \mathbb{V}_{A_t \sim \mu_t^{*,\text{PDIS}}} \left( \rho_t q_{\pi,t}(S_t, A_t) \mid S_t \right)$$
$$+ \mathbb{E}_{A_t \sim \mu_t^{*,\text{PDIS}}} \left[ \rho_t^2 \left( \mathbb{E}_{S_{t+1}} \left[ \mathbb{V} \left( G^{\text{PDIS}}(\tau_{t+1:T-1}^{\mu_{t+1:T-1}^{*,\text{PDIS}}}) \mid S_{t+1} \right) - \mathbb{V} \left( G^{\text{PDIS}}(\tau_{t+1:T-1}^{\mu_{t+1:T-1}^{*,b^*}}) \mid S_{t+1} \right) \mid S_t, A_t \right] \right) \mid S_t \right].$$
$$\text{(By (23))}$$

Thus, for $t = T - 1$,

$$\mathbb{V} \left( G^{\text{PDIS}}(\tau_{t:T-1}^{\mu_{t:T-1}^{*,\text{PDIS}}}) \mid S_t = s \right) - \mathbb{V} \left( G^{b^*}(\tau_{t:T-1}^{\mu_{t:T-1}^{*,b^*}}) \mid S_t = s \right)$$
$$= \mathbb{V}_{A_t \sim \mu_t^{*,\text{PDIS}}} \left( \rho_t q_{\pi,t}(S_t, A_t) \mid S_t \right). \qquad \text{(By (28) and (34))}$$

For $t \in [T - 2]$,

$$\mathbb{V} \left( G^{\text{PDIS}}(\tau_{t:T-1}^{\mu_{t:T-1}^{*,\text{PDIS}}}) \mid S_t = s \right) - \mathbb{V} \left( G^{b^*}(\tau_{t:T-1}^{\mu_{t:T-1}^{*,b^*}}) \mid S_t = s \right)$$
$$= \mathbb{V}_{A_t \sim \mu_t^{*,\text{PDIS}}} \left( \rho_t q_{\pi,t}(S_t, A_t) \mid S_t \right)$$
$$+ \mathbb{E}_{A_t \sim \mu_t^{*,\text{PDIS}}} \left[ \rho_t^2 \left( \mathbb{E}_{S_{t+1}} \left[ \mathbb{V} \left( G^{\text{PDIS}}(\tau_{t+1:T-1}^{\mu_{t+1:T-1}^{*,\text{PDIS}}}) \mid S_{t+1} \right) - \mathbb{V} \left( G^{\text{PDIS}}(\tau_{t+1:T-1}^{\mu_{t+1:T-1}^{*,b^*}}) \mid S_{t+1} \right) \mid S_t, A_t \right] \right) \mid S_t \right].$$
$$\text{(By (29)and (35))}$$

The proof of $\forall t, s, \delta_t^{\text{ODI, ours}}(s) \geq 0$ is similar to (33) and is omitted.

$\square$

### A.7 PROOF OF THEOREM 6

*Proof.* We begin the proof by manipulating the variance of $G^{b^*}(\tau_{t:T-1}^{\pi_{t:T-1}})$. $\forall s$, for $t = T - 1$,

$$\mathbb{V} \left( G^{b^*}(\tau_{t:T-1}^{\pi_{t:T-1}}) \mid S_t = s \right) \tag{36}$$
$$= \mathbb{E}_{A_t \sim \pi_t} \left[ [q_{\pi,t}(S_t, A_t) - b_t^*(S_t, A_t)]^2 \mid S_t \right] - [v_{\pi,t}(S_t) - \bar{b}_t^*(S_t)]^2 \quad \text{(Lemma 5 and on-policy)}$$
$$= 0 \qquad \text{(Definition of } b^* \text{ (15))}$$
$$= \mathbb{E}_{A_t \sim \pi_t} \left[ u_t^{b^*}(S_t, A_t) \mid S_t \right]. \qquad \text{(By (27))}$$

For $t \in [T - 2]$,

$$\mathbb{V} \left( G^{b^*}(\tau_{t:T-1}^{\pi_{t:T-1}}) \mid S_t = s \right) \tag{37}$$

$$=\mathbb{E}_{A_t\sim\pi_t}\left[\mathbb{E}_{S_{t+1}}\left[\mathbb{V}\left(G^{b^*}(\tau_{t+1:T-1}^{\pi_{t+1:T-1}})\mid S_{t+1}\right)\mid S_t,A_t\right]+\nu_t(S_t,A_t)+[q_{\pi,t}(S_t,A_t)-b_t(S_t,A_t)]^2\mid S_t\right]$$

$$-[v_{\pi,t}(S_t)-\bar{b}_t(S_t)]^2 \qquad\qquad\qquad\text{(By Lemma 5)}$$

$$=\mathbb{E}_{A_t\sim\pi_t}\left[\mathbb{E}_{S_{t+1}}\left[\mathbb{V}\left(G^{b^*}(\tau_{t+1:T-1}^{\pi_{t+1:T-1}})\mid S_{t+1}\right)\mid S_t,A_t\right]+\nu_t(S_t,A_t)\mid S_t\right]$$

$$+\mathbb{V}_{A_t\sim\pi_t}\left([q_{\pi,t}(S_t,A_t)-b_t^*(S_t,A_t)]\mid S_t\right) \qquad\qquad\text{(Lemma 6)}$$

$$=\mathbb{E}_{A_t\sim\pi_t}\left[\mathbb{E}_{S_{t+1}}\left[\mathbb{V}\left(G^{b^*}(\tau_{t+1:T-1}^{\pi_{t+1:T-1}})\mid S_{t+1}\right)\mid S_t,A_t\right]+\nu_t(S_t,A_t)\mid S_t\right] \qquad\text{(By (15))}$$

$$=\mathbb{E}_{A_t\sim\pi_t}\left[\mathbb{E}_{S_{t+1}}\left[\mathbb{V}\left(G^{b^*}(\tau_{t+1:T-1}^{\mu_{t+1:T-1}^{*,b^*}})\mid S_{t+1}\right)\mid S_t,A_t\right]+\nu_t(S_t,A_t)\mid S_t\right]$$

$$+\mathbb{E}_{A_t\sim\pi_t}\left[\mathbb{E}_{S_{t+1}}\left[\mathbb{V}\left(G^{b^*}(\tau_{t+1:T-1}^{\pi_{t+1:T-1}})\mid S_{t+1}\right)-\mathbb{V}\left(G^{b^*}(\tau_{t+1:T-1}^{\mu_{t+1:T-1}^{*,b^*}})\mid S_{t+1}\right)\mid S_t,A_t\right]\mid S_t\right]$$

$$=\mathbb{E}_{A_t\sim\pi_t}\left[u_t^{b^*}(S_t,A_t)\mid S_t\right]$$

$$+\mathbb{E}_{A_t\sim\pi_t}\left[\mathbb{E}_{S_{t+1}}\left[\mathbb{V}\left(G^{b^*}(\tau_{t+1:T-1}^{\pi_{t+1:T-1}})\mid S_{t+1}\right)-\mathbb{V}\left(G^{b^*}(\tau_{t+1:T-1}^{\mu_{t+1:T-1}^{*,b^*}})\mid S_{t+1}\right)\mid S_t,A_t\right]\mid S_t\right].$$
$$\text{(By (27))}$$

Thus, for $t=T-1$, their difference is

$$\mathbb{V}\left(G^{b^*}(\tau_{t:T-1}^{\pi_{t:T-1}})\mid S_t=s\right)-\mathbb{V}\left(G^{b^*}(\tau_{t:T-1}^{\mu_{t:T-1}^{*,b^*}})\mid S_t=s\right)$$

$$=\mathbb{E}_{A_t\sim\pi_t}\left[u_t^{b^*}(S_t,A_t)\mid S_t\right]-\mathbb{E}_{A_t\sim\pi_t}\left[\sqrt{u_t^{b^*}(S_t,A_t)}\mid S_t\right]^2 \qquad\text{(By (28) and (36))}$$

$$=\mathbb{V}_{A_t\sim\pi_t}\left(\sqrt{u_t^{b^*}(S_t,A_t)}\mid S_t\right).$$

For $t\in[T-2]$,

$$\mathbb{V}\left(G^{b^*}(\tau_{t:T-1}^{\pi_{t:T-1}})\mid S_t=s\right)-\mathbb{V}\left(G^{b^*}(\tau_{t:T-1}^{\mu_{t:T-1}^{*,b^*}})\mid S_t=s\right)$$

$$=\mathbb{E}_{A_t\sim\pi_t}\left[u_t^{b^*}(S_t,A_t)\mid S_t\right]-\mathbb{E}_{A_t\sim\pi_t}\left[\sqrt{u_t^{b^*}(S_t,A_t)}\mid S_t\right]^2$$

$$+\mathbb{E}_{A_t\sim\pi_t}\left[\mathbb{E}_{S_{t+1}}\left[\mathbb{V}\left(G^{b^*}(\tau_{t+1:T-1}^{\pi_{t+1:T-1}})\mid S_{t+1}\right)-\mathbb{V}\left(G^{b^*}(\tau_{t+1:T-1}^{\mu_{t+1:T-1}^{*,b^*}})\mid S_{t+1}\right)\mid S_t,A_t\right]\mid S_t\right]$$
$$\text{(By (29) and (37))}$$

$$=\mathbb{V}_{A_t\sim\pi_t}\left(\sqrt{u_t^{b^*}(S_t,A_t)}\mid S_t\right)$$

$$+\mathbb{E}_{A_t\sim\pi_t}\left[\mathbb{E}_{S_{t+1}}\left[\mathbb{V}\left(G^{b^*}(\tau_{t+1:T-1}^{\pi_{t+1:T-1}})\mid S_{t+1}\right)-\mathbb{V}\left(G^{b^*}(\tau_{t+1:T-1}^{\mu_{t+1:T-1}^{*,b^*}})\mid S_{t+1}\right)\mid S_t,A_t\right]\mid S_t\right].$$

The proof of $\forall t,s,\delta_t^{\text{DR, ours}}(s)\geq 0$ is similar to (33) and is omitted.

$\square$

## A.8 PROOF OF LEMMA 3

**Lemma 3** (Recursive form of $u$). *With $b=b^*$, when $t=T-1$, $\forall s,a$, $u_{\pi,t}(s,a)=0$, when $t\in[T-2]$, $\forall s,a$,*

$$u_{\pi,t}(s,a)=\nu_{\pi,t}(s,a)+\sum_{s',a'}\rho_{t+1}p(s'|s,a)\pi_{t+1}(a'|s')u_{\pi,t+1}(s',a').$$

*Proof.* When $t = T - 1$, $\forall s, a$,

$$u_{\pi,t}(s,a)$$
$$= (q_{\pi,t}(s,a) - b_t^*(s,a))^2 + \nu_{\pi,t}(s,a) \qquad \text{(By (27))}$$
$$= 0. \qquad \text{(By (15))}$$

When $t \in [T - 2]$, $\forall s, a$,

$$u_{\pi,t}(s,a)$$
$$= \nu_{\pi,t}(s,a) + \sum_{s'} p(s'|s,a) \mathbb{V}\left(G^b(\tau_{t+1:T-1}^{\mu_{t+1:T-1}^*}) \mid S_{t+1} = s'\right) \qquad \text{(By (27))}$$

$$= \nu_{\pi,t}(s,a) + \sum_{s'} p(s'|s,a) \left[ \mathbb{E}_{A_{t+1} \sim \mu_{t+1}^*} \left[ \rho_{t+1}^2 u_{\pi,t+1}(S_{t+1}, A_{t+1}) \mid S_{t+1} = s'\right]\right.$$
$$\left. - [v_{\pi,t+1}(s') - \bar{b}_{t+1}^*(s')]^2 \right] \qquad \text{(Lemma 5)}$$

$$= \nu_{\pi,t}(s,a) + \sum_{s'} p(s'|s,a) \mathbb{E}_{A_{t+1} \sim \mu_{t+1}^*} \left[ \rho_{t+1}^2 u_{\pi,t+1}(S_{t+1}, A_{t+1}) \mid S_{t+1} = s'\right] \qquad \text{(By (15))}$$

$$= \nu_{\pi,t}(s,a) + \sum_{s',a'} \rho_{t+1} p(s'|s,a) \pi_{t+1}(a'|s') u_{\pi,t+1}(s', a').$$

$\square$

## B    EXPERIMENT DETAILS

We utilize the behavior policy-agnostic offline learning setting (Nachum et al., 2019), in which the offline data consists of $\{(t_i, s_i, a_i, r_i, s_i')\}_{i=1}^m$, with $m$ previously logged data tuples. Those tuples can be generated by one or multiple behavior policies, regardless of whether these policies are known or unknown, and they are not required to form a complete trajectory. In the $i$-th data tuple, $t_i$ represents the time step, $s_i$ is the state at time step $t_i$, $a_i$ is the action executed, $r_i$ is the sampled reward, and $s_i'$ is the successor state.

In this paper, we first learn the action-value function $q_{\pi,t}$ from offline data using Fitted Q-Evaluation algorithms (FQE, Le et al. (2019)), but our method can integrate state-of-the-art offline policy evaluation techniques. Notably, Fitted Q-Evaluation (FQE, Le et al. (2019)) is a different algorithm from Fitted Q-Improvement (FQI). Fitted Q-Evaluation is not prone to overestimate the action-value function $q_{\pi,t}$ because Fitted Q-Evaluation does not have any max operator and does not change the policy.

Then, by the following derivation

$$\nu_{\pi,t}(s,a)$$
$$= \mathbb{V}_{S_{t+1}} \left( v_{\pi,t+1}(S_{t+1}) \mid S_t = s, A_t = a \right) \qquad \text{(By (11))}$$
$$= \mathbb{E}_{S_{t+1}} \left[ v_{\pi,t+1}(S_{t+1})^2 \mid S_t = s, A_t = a \right] - \mathbb{E}_{S_{t+1}} \left[ v_{\pi,t+1}(S_{t+1}) \mid S_t = s, A_t = a \right]^2$$
$$= \mathbb{E}_{S_{t+1}} \left[ v_{\pi,t+1}(S_{t+1})^2 \mid S_t = s, A_t = a \right] - (q_{\pi,t}(s,a) - r(s,a))^2, \qquad (38)$$

the first term is an expectation of $S_{t+1}$. Because we have $(t_i, s_i, a_i, r_i, s_i')$ data tuples, we construct $\nu$ using $s_i'$ in $(t_i, s_i, a_i, r_i, s_i')$ data tuples as the sample of the first term and compute the rest quantity using the learned action-value function $q_{\pi,t}$ and reward data $r_i$. Therefore, we construct $\mathcal{D}_\nu \doteq \{(t_i, s_i, a_i, \nu_i, s_i')\}_{i=1}^m$. Finally, by passing data tuples in $\mathcal{D}_\nu$ from $t = T - 1$ to $0$, we fit the function $u_{\pi,t}$ using FQE in a dynamic programming way with respect to the recursive form of $u_{\pi,t}$ derived in Lemma 3. For each time step, we take a copy of the neural network as an approximation of function $u_{\pi,t}$ at time step $t$. After learning the functions $u_{\pi,t}$ and $q_{\pi,t}$, we return the learned behavior policy $\mu_t^*(a|s) \propto \pi_t(a|s) \sqrt{u_{\pi,t}(s,a)}$ and the learned baseline function $b_t^*(s,a) = q_{\pi,t}(s,a)$. The pseudocode of this procedure is presented in Algorithm 1.

### B.1    GRIDWORLD

For a Gridworld with size $n$, we set its width, height, and time horizon all to be $n$. The number of states in this Gridworld environment scales cubically with $n$, offering a suitable tool to test algorithm

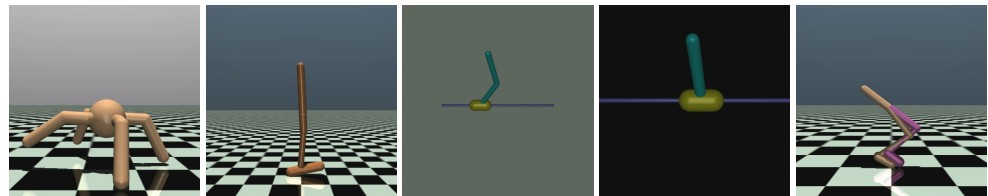

Figure 3: MuJoCo robot simulation tasks (Todorov et al., 2012). The pictures are adapted from (Liu and Zhang, 2024). Environments from the left to the right are Ant, Hopper, InvertedDoublePendulum, InvertedPendulum, and Walker.

scalability. We choose Gridworld with $n^3 = 1,000$ and $n^3 = 27,000$, the largest Gridworld environment tested among related works (Jiang and Li, 2016; Hanna et al., 2017; Liu and Zhang, 2024). There are four possible actions: left, right, up, and down. After the agent takes an action, it has a probability of 0.9 to move accordingly and a probability of 0.1 to move uniformly at random. If runs into a boundary, the agent stays in its current location. The reward function $r(s, a)$ is randomly generated. We consider 30 randomly generated target policies. We generate the ground truth policy performance using the on-policy Monte Carlo method, running each target policy for $10^6$ episodes. We test two different environment sizes of the Gridworld, one with $1,000$ states and $27,000$ states. The offline dataset of both environments contains $1,000$ episodes generated by a set of random policies. To learn functions $q_{\pi,t}$ and $u_{\pi,t}$, we split the offline data into a training set and a test set. We tune all hyperparameters offline based on Fitted Q-learning loss on the test set. We choose a one-hidden-layer neural network and test the neural network size with $[64, 128, 256]$ and choose 64 as the final size. We test the learning rate for Adam optimizer with $[1e^{-5}, 1e^{-4}, 1e^{-3}, 1e^{-2}]$ and choose to use the default learning rate $1e^{-3}$ as learning rate for Adam optimizer (Kingma and Ba, 2015). All benchmark algorithms are learned using their reported hyperparameters (Jiang and Li, 2016; Liu and Zhang, 2024). Each policy has 30 independent runs, resulting in $30 \cdot 30 = 900$ total runs. Therefore, each curve in Figure 1 is averaged from 900 different runs over a wide range of policies, showing a strong statistical significance.

## B.2 MuJoCo

MuJoCo is a physics engine containing various stochastic environments, where the goal is to control a robot to achieve different behaviors such as walking, jumping, and balancing. Environments in Figure 3 from the left to the right are Ant, Hopper, InvertedDoublePendulum, InvertedPendulum, and Walker. We construct 30 policies in each environment (resulting a total of 150 policies), incorporating a wide range of performance generated by the proximal policy optimization (PPO) algorithm (Schulman et al., 2017). We use the the default PPO implementation in Huang et al. (2022). We set each MuJoCo environment to have a fixed time horizon 100 in OpenAI Gymnasium (Towers et al., 2024). As our methods are designed for discrete action space, we discretize the first dimension of MuJoCo action space. The remaining dimensions are controlled by the PPO policies, and they are deemed as part of the environment. The offline dataset of each environment contains $1,000$ episodes generated by a set of policies with various performances. Functions $q_{\pi,t}$ and $u_{\pi,t}$ are learned the same way as in Gridworld environments. Our algorithm is robust on hyperparameters. All hyperparameters in Algorithm 1 are tuned offline and are the same across all MuJoCo and Gridworld experiments. Each policy in MuJoCo also has 30 independent runs, resulting in $30 \cdot 30 = 900$ total runs. Therefore, each curve in Figure 2 and each number in Table 2 are averaged from 900 different runs over a wide range of policies indicating strong statistical significance.

