# OpenReview forum: "Doubly Optimal Policy Evaluation for Reinforcement Learning"
_ICLR.cc/2025/Conference — ICLR 2025 Poster_

### Official Review · Reviewer_c89B · 2024-10-28

**Soundness:** 3
**Presentation:** 3
**Contribution:** 3
**Rating:** 8
**Confidence:** 4

**Summary:**

This paper studies the problem of efficient policy evaluation: designing a behavioral policy to collect data in order to estimate a target policy s.t. It requires as little as possible policy rollouts in order to evaluate the target policy performance for a given confidence level.

The authors cast this problem as a double step minimization problem of the variance of the importance sampling corrected reward traces (termed the G^PDIS), where the components to minimize are the behavioral policy \mu and a baseline function b that can reduce the overall variance of G^PDIS, while maintaining unbiasedness s.t. E[G^PDIS] converges to the target policy performance.

The authors develop an algorithm, named DOPT, inspired by two established methods for policy evaluation: (1) DR which uses the optimal baseline function and (2) ODI which do not take into account a baseline function, but solves for the optimal behavioral policy that minimizes the variance of G^PDIS (again without inserting a baseline function).

The authors show theoretically that if one adds both the optimal baseline function and solves for the optimal policy given the baseline we obtain a stronger estimator that minimizes the variance terms with respect to both DR and ODI (which are both superior over the on-policy case, where one uses rollouts from the target policy to approximate its performance).

The paper then continues with the practical implementation of how to estimate the theoretically justified behavioral policy and it concludes with an experiment section where it is shown that DOPT outperforms both ON (on-policy), DR and ODI in terms of evaluation steps with respect to the relative error (the difference between the estimated performance to the real performance scaled by this difference for on-policy evaluation after one episode). The experiments are executed in a synthetic grid-world and in the Mujoco physics simulator.

**Strengths:**

The related work section is comprehensive, contextualizing the paper within current literature and algorithms. The paper is generally clear and accessible, a crucial feature given the depth of theoretical content. The theoretical analysis rigorously establishes DOPT's optimality over on-policy, DR, and ODI baselines.

**Weaknesses:**

While optimal theoretically, in practice DOPT requires off-policy estimation of (1) the Q-function q(s,a), (2) the next state value variance \ni(s, a) and (3) the amplification factor u(s,a) (how much should the probability of an action be amplified for a given s,a pair). Specifically, u is learned based on a Bellman operator with \ni as a reward and with an important sampling correction term (Lemma 3). Both \ni and u are not required by previous schemes: DR and ODI and both can introduce errors. Specifically, u which uses Bellman operator that can potentially add bootstrapping errors and importance sampling term with high variance. Although empirical results support DOPT’s advantage, further analysis of these considerations is warranted for a complete evaluation. The authors should do a better job in analyzing and presenting these considerations.

**Questions:**

1. In your formulation your assume that Q_{\pi} is learned from the offline data, one would expect that a basic estimator of V(S0) in this case would be \sum_a \pi(a|S0) Q(a, S0) you should explain why it is not an appropriate estimator and add it to some of your empirical evaluations.

2. Based on your arguments in Section 4 (Lines 190-230) you require unbiasedness in every state not just s0, how does it reconcile with your argument that you enlarge the state space?

3. Figure 1 shows similar patterns for sample counts of 1000 and 27000. Can the authors clarify why similar accuracy requires the same number of samples for differently scaled problems?

4. Including a background on DR and ODI in the background section could enhance reader comprehension, as these are central to the analytical comparisons made.

5. Details on estimating \ni in practical settings should be incorporated into the main text. Without this information, readers may struggle to understand the algorithm’s practical execution.

---

> ### Author Response · Authors · 2024-11-20
>
> Many thanks for the encouraging feedback and detailed review. Your comments truly highlight the core strength of this work: combining rigorous theoretical results with state-of-the-art empirical performance.
>
>
> >In your formulation your assume that Q_{\pi} is learned from the offline data, one would expect that a basic estimator of V(S0) in this case would be \sum_a \pi(a|S0) Q(a, S0) you should explain why it is not an appropriate estimator and add it to some of your empirical evaluations.
>
> Thank you for your question! We want to clarify that in the setting we consider, the goal is to obtain an **unbiased** estimator. The $q$ value computed from the offline data is **biased**, as clarified in by Levein et al, 2020, “Offline Reinforcement Learning: Tutorial, Review, and Perspectives on Open Problems”. There is no way to quantify this bias in practice. By contrast, our method is **theoretically guaranteed to give an unbiased estimation** regardless of the offline data we use (Lemma 2).
>
> > Based on your arguments in Section 4 (Lines 190-230) you require unbiasedness in every state not just s0, how does it reconcile with your argument that you enlarge the state space?
>
> Thank you again for your insightful question. To begin with, we have enlarged the search space for *policies* instead of *states*. Specifically, in line 190-200, we presents the traditional wisdom which searches the unbiased behavior policies in **a smaller policy space**,
>
> $\Lambda^- = \{\mu \mid
> \forall t, s, a, \mu_t(a|s) = 0 \implies \pi_t(a|s) = 0\}$.
>
> In comparison, we search for the behavior policies in a larger space while still ensuring the **unbiasedness** (Lemma 2):
>
> $\Lambda = {\mu \mid \forall t, s, a, \mu_t(a|s) = 0 \implies \pi_t(a|s)u_{\pi, t}(s, a)  = 0}$.
>
> A detailed proof of $\Lambda^-\subseteq \Lambda$ is provided in our Appendix A.1. The boardness of our policy search space enables us to find a better variance-minimizing behavior policy compared with the traditional wisdom.
>
> >Figure 1 shows similar patterns for sample counts of 1000 and 27000. Can the authors clarify why similar accuracy requires the same number of samples for differently scaled problems?
>
> Thank you for your question. Let’s clarify point-by-point.
>
> **1. Larger Gridworld uses larger data size**
>
> The x-axis of Figure 1 represents the number of online **episodes**, as specified in line 476-477.
> To achieve the same *relative error*, the data size needed for the larger Gridworld is **also larger**. Specifically, in Girdworld with size 1000, the episode length is 10; while in Girdworld with size 27,000, the episode length is 30. Thus, under the same number of episodes, the data size is bigger for Girdworld with size 27,000 because of the longer episode length.
>
> **2.  Y-axis as relative error**
>
> The y-axis on Figure 1 represents the *relative error*. As described in line 478-479, *relative error* is the error of each estimator normalized by the estimation error of the on-policy Monte Carlo estimator after the first episode. We use relative error beacause we believe it is a more direct representation of the variance reduction across different enviornments. This representation is also adopted by the ODI paper (Liu and Zhang, ICML 2024). Therefore, the same number of online samples does not lead to similar absolute error across differen Gridworlds.
>
> **3. Difference between DR and ODI**
>
> Besides, there is another notable difference in the results from Gridworld with different sizes. As shown in *Table 1* of our paper, in Girdworld with size 1,000, DR outperforms ODI; while in Girdworld with size 27,000, ODI outperforms DR. Additionally, our method outperforms both ODI and DR by a great margin. **This is because our method considers variance reduction in both data-collecting and data-processing phases.**
>
>
> >Including a background on DR and ODI in the background section could enhance reader comprehension, as these are central to the analytical comparisons made.
>
> Thank you for your suggestion! We have taken this to heart. We have spent our Section 5 comparing the variance reduction property of our method with ODI and DR (Theorem 5 & 6). We have also provided comprehensive discussions on these two works in the related work section too.
>
> >Details on estimating \ni in practical settings should be incorporated into the main text. Without this information, readers may struggle to understand the algorithm’s practical execution.
>
> Many thanks for your constructive comment! We have added more detailes in estimating $\nu$ in our main text (line 435-436).
>
> Thank you again for your positive review, and please let us know if we have addressed your questions!

---

### Official Review · Reviewer_EiDc · 2024-11-01

**Soundness:** 2
**Presentation:** 3
**Contribution:** 3
**Rating:** 5
**Confidence:** 3

**Summary:**

This paper aims to enhance the sample efficiency of online reinforcement learning algorithms by reducing the variance of estimation of the total rewards. To achieve this, the authors introduce a bi-level optimization method with both optimal data-collecting policy and data-processing baseline. The paper provides detailed proof showing a guaranteed lower variance of the proposed method than those from the state-of-the-art methods. The empirical results on MiniGrid and tasks from the Mujoco environment conducted in this paper further demonstrate the effectiveness of the proposed method, outperforming existing methods in these tasks.

**Strengths:**

+ The paper is quite well-written and easy to follow. The motivation and the key idea of the method are clearly presented. The differences between the proposed method and prior work have been clearly discussed in the related work section.
+ The authors provide a detailed and comprehensive theoretical proof to support the proposed method, though there seem to be some errors in the proof of theorem 1.
+ The experiments show performance improvement over existing methods.

**Weaknesses:**

- A potential error in the proof:

In equation (23), the second equal sign does not make sense. By equation (12) that

${\mu}^{*}_t (a|s) \propto \pi _ t (a|s) \sqrt{u _ { \pi, t} (s,a)}$.

We should have

$\sum_{a} \frac{\pi _ t(a|S_t)^2}{\mu^{*}_t (a|S_t)} u _ {\pi, t}(S_t,a)$

$= \sum_{a} \pi _ t (a|S_t)\sqrt{u _ {\pi, t} (S_t, a)}$

However, in the paper the author seems to give the wrong result as

$\sum_{a} \frac{\pi _ t(a|S_t)^2}{\mu^{*}_t (a|S_t)} u _ {\pi, t}(S_t,a)$

$=\sum_{a}\pi_t(a|S_t)\sqrt{u _ {\pi, t}(S_t, a)} \sum_{b}\pi_t(S_t, b)\sqrt{u _ {\pi, t}(S_t, b)}$.

Unless I'm missing something, this seems to be an error that may invalidate the optimality of the proposed behavior policy, as the results is used in subsequent analysis and proofs. The proof also contains typos and minor issues, such as the following. All of these cast doubt on the soundness of the analysis.

After the second equal sign in equation (18), there's an extra ")" after $\bar{b_t}(S_t)$;

At the start of equation (20), $\nu_t(S_t, A_t) =$ should be removed;

In page 17, the $\mu^{*}_t$ after the third equal sign should be $\mu_t$.

- Figure 3 looks exactly the same as the Figure 3 in the following paper [1], however without citing the source. This is a major concern.

[1]Liu, S. and Zhang, S. (2024). Efficient policy evaluation with offline data informed behavior policy design. In Proceedings of the International Conference on Machine Learning.

- While the experiments show good improvement of baseline algorithms, I would suggest the experiments be expanded to more environments such as Atari (RL with visual input) and more tasks in MuJoCo, where the sampling are generally less efficient.

**Questions:**

1. Please address my concerns in the weakness.

2. According to algorithm 1, will the offline dataset affect the performance? In the experiments these offline datasets are collected by random policies. I wonder what if it is collected by an expert policy or other more sophisticated behavior policies, like in most RL settings? This is important to demonstrate the applicability of the proposed method on datasets with different expertise levels.

---

> ### Author Response · Authors · 2024-11-20
>
> Thank you for your time and comments. As you pointed out, our method is theoretically guaranteed and empirically proven to achieve lower variance than previous best performing methods.
>
> ### For weaknesses:
> >A potential error in the proof… Unless I'm missing something, this seems to be an error that may invalidate the optimality of the proposed behavior policy, as the results is used in subsequent analysis and proofs.
>
> Thank you for your feedback! However, we respectfully disagree with your assertion that this is an error. Let’s clarify it with a closer look.
>
> By equation (12),
> $\mu^{*}_t (a|s) \propto \pi _ t (a|s) \sqrt{u _ { \pi, t} (s,a)}$
> where the $\propto $ means “proportional to”. This notation is widely used, for example, by Sutton and Barto (2018), *Reinforcement Learning: An Introduction*, in their Chapter 13.
>
> That is,
> $\mu^{*}_t (a|s) = \frac{\pi _ t(a|s) \sqrt{u _ { \pi, t} (s,a)}}{\sum_b\pi_t(b|s)\sqrt{u _ { \pi, t}(s,b)}}$, as written in our paper line 272.
>
> Thus,
> $\sum_{a} \frac{\pi _ t(a|S_t)^2}{\mu^{*}_t (a|S_t)} u _ {\pi, t}(S_t,a)$
>
> $=\sum_{a} \frac{\pi _ t(a|S_t)^2}{\pi_t(a|s)\sqrt{u_{\pi,t}(s,a)}} \sum_b\pi_t(b|s)\sqrt{u_{\pi,t}(s,b)}u _ {\pi, t}(S_t,a)$
>
> $=\sum_{a}\pi_t(a|S_t)\sqrt{u _ {\pi, t}(S_t, a)} \sum_{b}\pi_t(S_t, b)\sqrt{u _ {\pi, t}(S_t, b)}$,
> which matches the proof in our paper.
>
> In your derivation, you might have treated "$\propto$" as "$=$". Thus, you obtained
>
> $\sum_{a} \frac{\pi _ t(a|S_t)^2}{\mu^{*}_t (a|S_t)} u _ {\pi, t}(S_t,a)$
>
> $=\sum_{a} \frac{\pi _ t(a|S_t)^2}{  \pi_t (a| S_t)  \sqrt{u_{\pi,t}(S_t,a)} }  u_{\pi,t}(S_t,a)$
>
> $= \sum_{a} \pi _ t (a|S_t)\sqrt{u _ {\pi, t} (S_t, a)}$.
>
> >After the second equal sign in equation (18), there's an extra ")" after $\bar{b}_t(S_t)$.
>
> Thank you for your detailed review. Indeed, we believe that the “)” is **necessary**. In (18), the term containing $\bar{b}_t(S_t)$ is:
>
> $V_{A_t}(\rho_t [q_{\pi,t}(S_t,A_t)-b_t(S_t,A_t)]+\bar{b}_t(S_t)\mid S_t)$.
> Here, the last “)” is for closing the variance term.
>
>
> >At the start of equation (20), $\nu_t(St,At)=$ should be removed.
>
> We sincerely appreciate your detailed review! We have deleted the extra term in our revision.
> >In page 17, the $\mu^*_t$ after the third equal sign should be $\mu_t$.
>
> Many thanks for pointing this out. We agree that this is a typo, and we have corrected it in the current paper.
> We assure you the last two typos are in the notation level of the appendix. Our results are checked by multiple fellows and are ensured to be correct. We thank you again for your careful review.
>
> >Figure 3 looks exactly the same as the Figure 3 in the following paper [1], however without citing the source.
>
> Thank you for your suggestion. Figure 3 is a visual demonstration of the MuJoCo tasks screenshotted from OpenAI Gym (Brockman et al., 2016). We have added the citation for this picture.
>
>
> >While the experiments show good improvement of baseline algorithms, I would suggest the experiments be expanded to more environments such as Atari (RL with visual input) and more tasks in MuJoCo, where the sampling are generally less efficient.
>
> Thank you for your advice. In fact, as shown in the following table, the environments we tested are among the **most complex environments** used in RL policy evaluation papers. Also, our method imposes weaker assumptions on the data.
>
> |   | Data to Learn $\mu $ | Parameterization of $\pi $ | Gridworld Size | Other Environments  |
> |-----------|--------------------------|-------------------------------|----------------|---------------------|
> | **Ours**  | offline data             | no assumption                 | 27,000         | MuJoCo robotics     |
> | (NeurIPS) Zhong et al. 2022   [1]    | online data              | needs to be known             | 1,600          | CartPole, Acrobot   |
> | (ICML) Hanna et al. 2017     [2] | online data              | needs to be known             | 1,600          | CartPole            |
> | (ICML) Liu and Zhang, 2024 [3]| offline data             | no assumption                 | 27,000         | MuJoCo robotics     |
>
> ### For questions:
>
> >According to algorithm 1, will the offline dataset affect the performance?
>
> Thank you for your questions. As with most offline RL approaches, the quality of offline data does impact the quality of the learned policy. This is **inherent** to offline RL and cannot be fully resolved, as pointed out by Levein et al, 2020, “Offline Reinforcement Learning: Tutorial, Review, and Perspectives on Open Problems”.

---

> ### Author Response · Authors · 2024-11-20
>
> >In the experiments these offline datasets are collected by random policies. I wonder what if it is collected by an expert policy or other more sophisticated behavior policies, like in most RL settings? This is important to demonstrate the applicability of the proposed method on datasets with different expertise levels.
>
> Many thanks for this constructive question!
> To answer your question, we made additional ablation studies to test the performance of our method when the offline data is generated by **an expert policy: the target policy**. This offline data set is considered to have better quality because it contains more state-action pairs that are frequently visited by the target policy. As for the other columns, the offline data is generated by 30 different policies with **various performances**. Both types of data sets contain $1000$ episodes.
>
> |               | On-policy MC | **Ours-Expert** | **Ours** | ODI   | DR    |
> |---------------|--------------|------------------|----------|-------|-------|
> | Ant       | 1.000        | **0.413**        | 0.493    | 0.811 | 0.636 |
> | Hopper    | 1.000        | **0.304**        | 0.372    | 0.542 | 0.583 |
> | I. D. Pendulum | 1.000   | **0.358**        | 0.427    | 0.724 | 0.652 |
> | I. Pendulum    | 1.000   | **0.203**        | 0.226    | 0.351 | 0.440 |
> | Walker    | 1.000        | **0.427**        | 0.475    | 0.696 | 0.656 |
>
> *Table: Relative variance of estimators on MuJoCo. The relative variance is defined as the variance of each estimator divided by the variance of the on-policy Monte Carlo estimator. Numbers are averaged over 900 independent runs (30 target policies, each having 30 independent runs).*
>
> |               | On-policy MC | **Ours-Expert** | **Ours** | ODI   | DR    | Saved Episodes Percentage      |
> |---------------|--------------|------------------|----------|-------|-------|---------------------------------|
> | Ant       | 1000         | **416**         | 492      | 810   | 636   | **50.8% – 58.4%**              |
> | Hopper    | 1000         | **306**         | 372      | 544   | 582   | **62.8% – 69.4%**              |
> | I. D. Pendulum | 1000   | **358**         | 426      | 727   | 651   | **57.4% – 64.2%**              |
> | I. Pendulum    | 1000   | **204**         | 225      | 356   | 439   | **77.5% – 79.6%**              |
> | Walker    | 1000         | **429**         | 475      | 705   | 658   | **52.5% – 57.1%**              |
> *Table: Episodes needed to achieve the same estimation accuracy that on-policy Monte Carlo achieves with 1000 episodes on MuJoCo.*
>
> [Figure (link)](https://drive.google.com/file/d/1jPVyVcD4IU5MUosNE8SNo2mNEYAeG41t/view?usp=sharing): *Results on MuJoCo. Each curve is averaged over 900 independent runs. Shaded regions denote standard errors and are invisible for some curves because they are too small.*
>
> As shown in the results, data generated by expert policy (target policies) improves performance, and our algorithm scales with data quality.
>
> The figure further shows that our method (with both expert and regular offline data set) **outperforms all the other baselines by a large margin consistently**. This demonstrates the fact that the majority of improvement comes from the algorithmic side.
> Specifically, our method **reduces substantially more variance** compared with ODI (Liu and Zhang, ICML [3]) and DR (Jiang and Li, ICML [4]) because we leverage offline data to achieve **optimal online data collecting and data processing** for policy evaluation. This superiority is proved in our Theorem 5 and Theorem 6. Across different MuJoCo environments with different quality of offline data, **our method saves 50.8%-79.6% online samples**, achieving state-of-the-art performance.
>
> Thank you again for your review, and please let us know if we have addressed your comments and questions! We’ll be happy to discuss further.
>
> [1] (NeurIPS Zhong et al. 2022) "Robust On-Policy Sampling for Data-Efficient Policy Evaluation in Reinforcement Learning" (ROS)
>
> [2](ICML Hanna et al. 2017) "Data-Efficient Policy Evaluation Through Behavior Policy Search" (BPG)
>
> [3] (ICML Liu and Zhang, 2024) “Efficient Policy Evaluation with Offline Data Informed Behavior Policy Design” (ODI)
>
> [4] (ICML Jiang and Li, 2016) “Doubly Robust Off-Policy Value Evaluation For Reinforcement Learning” (DR)

---

> ### Author Response · Authors · 2024-11-25
>
> As the rebuttal phase is nearing its end, we wanted to kindly follow up to check if you had any additional feedback or comments for our paper. Your input would be greatly appreciated, and we are confident to discuss and address any concerns you may have.
>
> Thank you again for your time and effort in reviewing our work!

---

> > ### Author Response · Authors · 2024-12-01
> >
> > Thank you once again for the time and effort you have devoted to reviewing our paper. As we approach the end of the rebuttal phase, we wanted to kindly follow up to inquire if you have any further concerns or feedback that we could address.
> >
> > We greatly value your insights and are always happy to discuss further!

---

> > > ### Author Response · Authors · 2024-12-04
> > >
> > > We are sorry that we have not received any feedback from you during the rebuttal phase. Your major concern on our paper is from **misunderstanding** $\propto$ as $=$, which we have explained in our rebuttal, ensuring the correctness of our proof.
> > >
> > > We notice that your confidence level is **3**, which is the least confident level among all three reviewers, showing that "it is possible that you did not understand some parts of the submission or that you are unfamiliar with some pieces of related work. Math/other details were not carefully checked." Given that we have carefully addressed all concerns raised, including those from all other reviewers, we kindly request you to reconsider your evaluation and raise your rating if you find our clarifications satisfactory.
> > >
> > > We greatly appreciate your time and thoughtful consideration.

---

### Official Review · Reviewer_Zj5D · 2024-11-04

**Soundness:** 3
**Presentation:** 3
**Contribution:** 2
**Rating:** 6
**Confidence:** 4

**Summary:**

This paper studies the policy evaluation problem. Given a policy $\pi$ to evaluate, the authors adopted the classic importance-weighting estimator with baseline function. By optimizing the variance of this estimator, they derived an optimal behavior policy and baseline function. They show that the variance of the importance weighting estimator equipped with their optimal behavior policy and baseline function is smaller than three existing estimators including the on-policy estimator and the doubly robust estimator. Additionally, they implemented experiments on practical tasks to show the effectiveness of their doubly optimal evaluation method.

**Strengths:**

1. It is an interesting problem to study that to evaluate a given policy, what is the best behavior policy to collect samples.

**Weaknesses:**

The pipeline of their proposed method is that: First, calculating an optimal behavior policy and baseline function by solving an optimization problem; Second, using the derived behavior policy to collect samples and use importance-weight estimator on these samples.
1. However, in the first stage, in order to calculate the optimal behavior policy and baseline function, lots of samples are needed and I think the samples needed in this stage are much more than the second stage (since the definition of $u(s,a)$ needs lots of information e.g. the transition function). In this case, it is meaningless to reduce the variance of the estimator.
2. And the optimal baseline function is found to be the q-value function of the target policy. If we already figure out an optimal baseline function, i.e. the q-value of the target policy, then it is done. No need to do importance-weighting any more. Therefore, again it is meaningless to reduce the variance of the estimator.

The above is the biggest weakness. It is not clear what is achieved by their method theoretically or practically.

Besides, there are also other problems which make their theory fragile.

3. The set $\Lambda$ of feasible behavior policies seems weird to me. It contains behavior policy $\mu$ such that $\mu(a|s)=0$ while $\pi(a|s)>0$ ($q(s,a)=0$). In this case, the importance weighting estimator goes to infinity, let alone control the variance. One solution is to assume non-negative rewards, since with non-negative rewards $q(s,a)=0$ indicates zero rewards by playing $a$ at state $s$ which can make the importance weighting estimator be zero. However, in the paper, they don't discuss it and make any assumptions.

**Questions:**

My main question is what I have listed in the weaknesses part: what is the pipeline of the proposed method? If the pipeline is same as what I write in the weaknesses part, then what is meaning to reduce the variance of the estimator.

---

> ### Author Response · Authors · 2024-11-20
>
> Thank you for your time and comments! We believe we can address your comments and questions by detailed clarifications.
>
> ### For weaknesses:
> >However, in the first stage, in order to calculate the optimal behavior policy and baseline function, lots of samples are needed and I think the samples needed in this stage are much more than the second stage.
>
> **1. Why leverage offline data**
>
> We consider the setting where **offline data is much cheaper than online data**. This setting is widely studied for off-policy evaluation problems, as specified by the well-known ICML paper Jiang and Li, 2016, *Doubly Robust Off-Policy Value Evaluation For Reinforcement Learning*; and it is also a key motivation for offline RL, according to Levein et al, 2020, *Offline Reinforcement Learning: Tutorial, Review, and Perspectives on Open Problems*.
>
> Consider in most tech companies like TikTok, Google, and Amazon, there are large volumes of existing **offline data** obtained from previous algorithmic implementations. However, taking the advertisement recommendation systems widely used by these companies as an example, **excessive online evaluation** risks disrupting user experience and losing customers.
> Thus, these companies prioritize approaches that **maximize the utility of their offline data** while **minimizing online interactions**.
>
>
> **2.Existing methods**
>
> In most RL implementations, to avoid potential damage from bad target policies in the **online execution** phase, RL practitioners usually **use offline data to approximate $q$** a priori. This provides a preliminary, though **biased**, estimation of the target policy’s performance. In the existing best-performing approaches, **prior to performing policy evaluation**,  the DR method (Jiang and Li, 2016) learns $q$ from offline data to serve as a baseline function, and the ODI method [1] learns $q$ and $\hat{q}$ (an extended q-value they defined) from offline data to approximate their behavior policy. In short, their pipelines are:
>
> (1) Learn $q$ (and $\hat{q}$) from offline data.
>
> (2) Use the learned functions to perform online policy evaluation.
>
> However, since these two methods only focus on reducing variance in data collecting (ODI) or data processing (DR) phase, respectively, they have not effectively leveraged the offline data.
>
> **3. Our Superiority**
>
>    As clarified in our algorithm, we use **existing and cheap offline data** to learn the behavior policy.  The variance reduction property of our behavior policy can greatly reduce the samples needed in the **expensive online interaction**.
>
> Compared with the ODI and DR methods, which also need offline data to learn the $q$ function,  we innovatively reduce variance in both data collecting and data processing phases. Thus, it is **theoretically proved that our estimator achieves lower variance** than ODI and DR (Theorem 5 and Theorem 6). Empirically, our method outperforms the existing methods by a large margin across environments, achieving state-of-the-art performance in **saving online data**.
>
> In short, the pipeline of our method is:
>
> (1) Learn $q$ and $u$ from offline data.
>
> (2) Use the learned functions to perform online policy evaluation, **saving 50.8% to 77.5% online samples**.
>
> Please let us know if we have addressed your question about this setting! We’ll be happy to discuss further.

---

> ### Author Response · Authors · 2024-11-20
>
> >since the definition of u(s,a) needs lots of information e.g. the transition function.
>
> To learn the behavior policy $\mu$, we *do not* need to estimate the transition function. This is because, as written in our Algorithm 1, we use Fitted Q-Evaluation (Le et al. (2019)) to learn $q$ and $u$. Notably, Fitted Q-Evaluation is a model free algorithm that does not require estimating the transition function.
>
> >And the optimal baseline function is found to be the q-value function of the target policy. If we already figure out an optimal baseline function, i.e. the q-value of the target policy, then it is done. No need to do importance-weighting any more.
>
> **1.** **Biased Estimation from q**
>
> We want to clarify that in the setting we consider, the goal is to obtain an **unbiased** estimator. The $q$ value computed from the offline data is **biased**, as clarified in by Levein et al, 2020, *Offline Reinforcement Learning: Tutorial, Review, and Perspectives on Open Problems*.  There is no way to quantify this bias in practice. Thus, **it is not done** after we compute the $q$ value from offline data because it only gives biased estimation.
>
> **2.** **Leverage offline data while ensuring unbiased estimation**
>
> Bedies, as answered above for your first concern, using the $q$ value learned from the offline data to obtain an unbiased Monte Carlo estimator is a widely accepted norm. This strategy is adopted in the well-known ICML paper (Jiang and Li, 2016, *Doubly Robust Off-Policy Value Evaluation For Reinforcement Learning*) and the ODI paper [1] (also published on ICML).
>
> Compared with these two previously best-performing approaches, our method also gives inherently **unbiased estimation**. Moreover, since our method considers variance reduction in both data-collecting and data-processing phases, **it is theoretically guaranteed and empirically demonstrated to achieve lower variance than both of them**.
>
> >The set $\Lambda$ of feasible behavior policies seems weird to me. It contains behavior policy $\mu$ such that $\mu(a|s)=0$ while $\pi(a|s)>0$ $(q(s,a)=0)$. In this case, the importance weighting estimator goes to infinity, let alone control the variance.
>
> **1. Infinity term will not be sampled**
>
> We are actually drawing samples using the behavior policy $\mu$ when collecting data.
> In the example you give, the action $a$ that has $\mu(a|s) = 0, \pi(a|s) > 0$ **will never be sampled** from the behavior policy $\mu$  because $\mu(a|s) = 0$. Thus, the infinity importance sampling ratio will never appear in samples.
>
> **2. Superiority of our policy search space**
>
> Our policy searching space
>
> $\Lambda = \{ \mu \mid \forall t, s, a, \mu_t(a|s) = 0 \implies \pi_t(a|s)u_{\pi, t}(s, a)  = 0 \}$
>
>  is proved to guarantee **unbiasedness** (Lemma 2), and it is larger than the traditional search space
>
> $\Lambda^- = \{\mu \mid
> \forall t, s, a, \mu_t(a|s) = 0 \implies \pi_t(a|s) = 0\}$,  as proved in Lamma 1.
>
> A detailed proof of $\Lambda^-\subseteq \Lambda$ is provided in our Appendix A.1. The boardness of our policy search space enables us to find a **better variance-minimizing behavior policy** compared with the traditional wisdom.
>
> Thank you again for your review! And we would be glad to answer any further questions.
>
> [1]  (ICML Liu and Zhang, 2024) “Efficient Policy Evaluation with Offline Data Informed Behavior Policy Design” (ODI)

---

> ### Author Response · Authors · 2024-11-25
>
> As the rebuttal phase is nearing its end, we wanted to kindly follow up to check if you had any additional feedback or comments for our paper. Your input would be greatly appreciated, and we are confident to discuss and address any concerns you may have.
>
> Thank you again for your time and effort in reviewing our work!

---

> > ### Comment · Reviewer_Zj5D · 2024-11-26
> > **Main concern solved, increase score to 6**
> >
> > Thanks for the careful clarifications especially on the usage of offline dataset. I am convinced by the explanation that one can estimate a biased q function using large volume of offline data, then one builds the final unbiased estimator with low variance which can save online interactions. For another weakness, I am sorry that I threw out such a silly critique. It is obvious when $\mu(a|s)=0$, $(s, a)$ will not be sampled.
> >
> > The remaining minor questions I still have are:
> > 1. Why don't we need to estimate the transitions p(s'|s,a) when estimating u(s,a) based on the formula in Lemma 3?
> > 2. Different from other off-policy works where the behavior policy is a given safe policy, the optimal behavior policy in this paper is derived which can be dangerous to implement in some cases, right? I am curious if there is any backup. Have the authors ever considered this situation? Not a critique, just an interesting question.
> >
> > I will update my score to 6 based on the following considerations. My main concern (i.e. the huge amount data needed in the first stage of their pipeline) has been solved. The main claims made in this paper sound correct to me. At the same time, compared with ODI, the contribution of this papar is to incorporate an optimal baseline function which is not significant enough. Hence, I recommend a weak-acception.

---

> > > ### Author Response · Authors · 2024-11-29
> > >
> > > We sincerely appreciate your response and thorough review. And we are grateful for your turn-around!
> > >
> > > ### For questions:
> > > >Why don't we need to estimate the transitions p(s'|s,a) when estimating u(s,a) based on the formula in Lemma 3?
> > >
> > >
> > > By Lemma 3,
> > >
> > > $u_{\pi,t}(s,a)= \nu_{\pi, t}(s, a) + \sum_{s', a'} \rho_{t+1} p(s'|s, a) \pi_{t+1}(a'|s')  u_{\pi, t+1}(s', a')$
> > >
> > > $           = \nu_{\pi, t}(s, a) + E_{s’,a’}[ \rho_{t+1}   u_{\pi, t+1}(s', a')]$.
> > >
> > > Here, the second term is an expectation over the next state’s $u$. Because we are interested in the expectation, we can directly approximate the expectation from **offline samples** without explicitly estimating the transition probability $ p ( s’ | s , a) $. Specifically, as clarified in our Algorithm 1, we use Fitted Q-Evaluation (Le et al. (2019)) to learn $u$ iteratively. Notably, Fitted Q-Evaluation is a **model-free** algorithm that does not require estimating the transition function.
> > >
> > > >Different from other off-policy works where the behavior policy is a given safe policy, the optimal behavior policy in this paper is derived which can be dangerous to implement in some cases, right? I am curious if there is any backup. Have the authors ever considered this situation? Not a critique, just an interesting question.
> > >
> > > Thank you for your constructive question! We have taken this to heart.
> > >
> > > Our paper focuses on the doubly-optimal (optimal in both data-collecting and data-sampling phases) variance reduction for off-policy evaluation, and safety is beyond the scope of this work. Nevertheless, there is a concurrent work on safety-constrained off-policy evaluation [1], which focuses on the data-sampling phase. They **trade-off** variance reduction for safety in the design of their behavior policy, obtaining a safe but conservative estimator.
> > >
> > > Still, we believe that the safety issue in policy evaluation is worth exploring. The main technique in [1] can be directly implemented into our data collecting phase. For scenario where safety is a priority over the online data efficiency, we can use this constrained behavior policy to collect data.
> > >
> > > Please let us know if you have any further questions! We are happy to discuss.
> > >
> > > [1] (Chen et al, 2024) "Efficient Policy Evaluation with Safety Constraint for Reinforcement Learning"

---

### Meta-Review · Area_Chair_wY49 · 2024-12-24

**Metareview:**

In this paper, the authors propose a method to find a behavioral policy that can be used to estimate the value of a given target policy for a confidence level with small amount of data generation (policy rollouts). By optimizing the variance of the classic importance-weighting estimator with a baseline, they derive their behavior policy and baseline. They show that the variance of the importance weighting estimator with these behavior policy and baseline is smaller than three existing estimators, including the on-policy and doubly robust estimators. Finally, they empirically evaluate their proposed method.

Here are positives and negatives of the paper from the reviews:
(+) The paper is well-written. The motivation and key ideas are clearly presented. The related work is properly listed. The proposed results have been properly put in context with respect to the existing literature.
(+) The proposed method is well-supported by detailed and comprehensive theoretical results.
(+) The proposed method is empirically evaluated against existing methods.
(-) The extra complexities of the proposed method (compared to the existing methods) are not properly highlighted.

**Additional Comments On Reviewer Discussion:**

The authors were successfully addressed reviewers' questions, and thus, some of them raised their scores.

---

### Decision · Program_Chairs · 2025-01-22

Accept (Poster)